

# Carbon cycling in the North American coastal ocean: A synthesis

Katja Fennel[1], Simone Alin[2], Leticia Barbero[3], Wiley Evans[4], Timotheé Bourgeois[1], Sarah Cooley[5], John Dunne[6], Richard A. Feely[2], Jose Martin Hernandez-Ayon[7], Xinping Hu[8], Steven Lohrenz[9], Frank Muller-Karger[10], Raymond Najjar[11], Lisa Robbins[10], Elizabeth Shadwick[12], Samantha Siedlecki[13], Nadja Steiner[14], Adrienne Sutton[2], Daniela Turk[1], Penny Vlahos[13], and Zhaohui Aleck Wang[15]

[1]Department of Oceanography, Dalhousie University, 1355 Oxford Street, Halifax B3H 4R2, Nova Scotia, Canada
[2]NOAA Pacific Marine Environmental Laboratory
[3]NOAA Atlantic Oceanographic and Meteorological Laboratory
[4]Hakai Institute
[5]Ocean Conservancy
[6]NOAA Geophysical Fluid Dynamics Laboratory
[7]Autonomous University of Baja California
[8]Texas A&M University, Corpus Christi
[9]University of Massachusetts, Dartmouth
[10]University of South Florida
[11]Pennsylvania State University
[12]Virginia Institute of Marine Science
[13]University of Connecticut
[14]Department of Fisheries and Oceans Canada
[15]Woods Hole Oceanographic Institution

**Correspondence:** Katja Fennel (katja.fennel@dal.ca)

**Abstract.** A quantification of carbon fluxes in the coastal ocean and across its boundaries, specifically the air-sea, land-to-coastal-ocean and coastal-to-open-ocean interfaces, is important for assessing the current state and projecting future trends in ocean carbon uptake and coastal ocean acidification, but is currently a missing component of global carbon budgeting. This synthesis reviews recent progress in characterizing these carbon fluxes with focus on the North American coastal ocean.

Several observing networks and high-resolution regional models are now available. Recent efforts have focused primarily on quantifying net air-sea exchange of carbon dioxide ($CO_2$). Some studies have estimated other key fluxes, such as the exchange of organic and inorganic carbon between shelves and the open ocean. Available estimates of air-sea $CO_2$ flux, informed by more than a decade of observations, indicate that the North American margins act as a net sink for atmospheric $CO_2$. This net uptake is driven primarily by the high-latitude regions. The estimated magnitude of the net flux is $160 \pm 80$ Tg C y$^{-1}$ for the

North American Exclusive Economic Zone, a number that is not well constrained. The increasing concentration of inorganic carbon in coastal and open-ocean waters leads to ocean acidification. As a result conditions favouring dissolution of calcium carbonate occur regularly in subsurface coastal waters in the Arctic, which are naturally prone to low pH, and the North Pacific, where upwelling of deep, carbon-rich waters has intensified and, in combination with the uptake of anthropogenic carbon, leads to low seawater pH and aragonite saturation states during the upwelling season. Expanded monitoring and extension of existing

model capabilities are required to provide more reliable coastal carbon budgets, projections of future states of the coastal ocean, and quantification of anthropogenic carbon contributions.





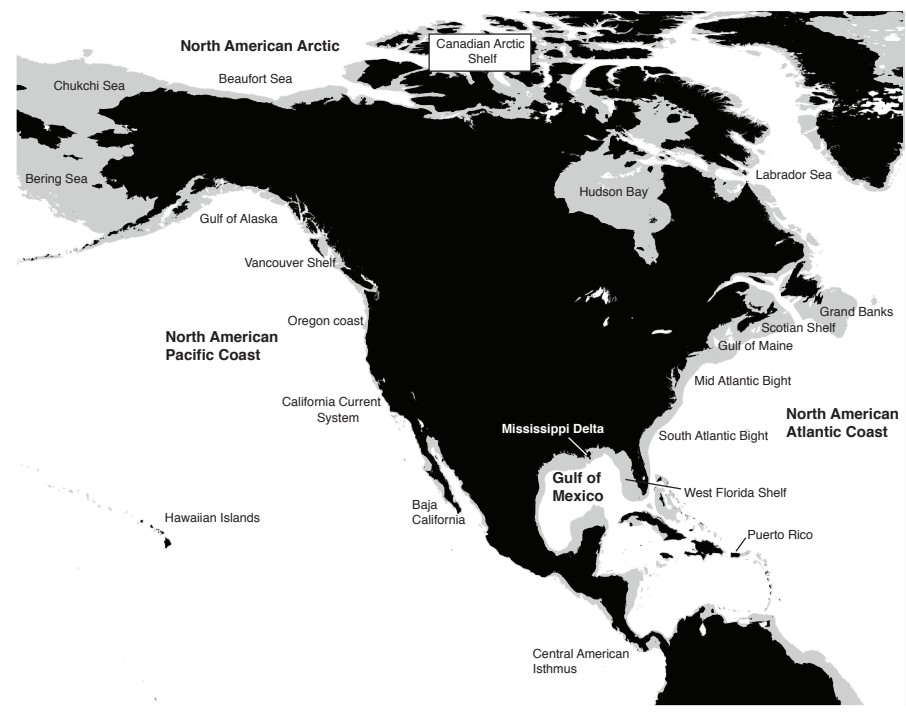

**Figure 1.** North American continent (in black) with shelf seas (in gray) defined as waters with bottom depths less than 200 m. The easternmost tip of Asia and northern part of South America are also shown in black.

## 1  Introduction

Along ocean margins, the atmospheric, terrestrial, sedimentary, and deep-ocean carbon reservoirs meet, resulting in quantitatively significant carbon exchanges. Anthropogenic activities lead to secular trends in these exchanges. The drivers underlying the secular trends include rising atmospheric carbon dioxide ($CO_2$) levels, climate-driven changes in atmospheric forcing (e.g., winds and heat fluxes), ocean circulation, the hydrological cycle (e.g., freshwater input from rivers), and changes in riverine and atmospheric nutrient inputs from agricultural activities and fossil fuel burning. The collective impact of these factors on carbon processing and exchanges along ocean margins is complex and difficult to quantify (Regnier et al., 2013).

This review aims to summarize recent findings with respect to coastal carbon uptake and ocean acidification for the ocean margins of North America (Figure 1) and was conducted as part of an assessment for the 2nd State of the Carbon Cycle Report (SOCCR-2). It builds on and extends several previous activities, including a report by the North American Continental Margins




Working Group (Hales et al., 2008), the First State of the Carbon Cycle Report (SOCCR-1; King et al., 2007), and activities within the North American coastal interim synthesis (Benway et al., 2016; Benway and Coble, 2014; Najjar et al., 2012).

A decade ago in SOCCR-1, Chavez et al. (2007) concluded that carbon fluxes for North American coastal margins were not well quantified because of insufficient observations and the complexity and highly localized spatial variability of coastal

carbon dynamics. The report was inconclusive as to whether North American coastal waters act as an overall source or sink of atmospheric $CO_2$. Here we revisit the question of whether the coastal ocean of North America takes up atmospheric $CO_2$ and subsequently exports it to the deep ocean, and discuss patterns and drivers of coastal ocean acidification. The first topic is relevant to overall quantification of the ocean's uptake of $CO_2$. The second is directly relevant to coastal ecosystem health, fisheries, and aquaculture.

Two different terms will be used here when referring to ocean margins: **coastal oceans**, defined here as non-estuarine waters within 200 nautical miles (370 km) of the coast, and **continental shelves**, which refer to the submerged margins of the continental plates, operationally defined here as regions with water depths shallower than 200 m (indicated in gray in Figure 1). Although the two definitions overlap, there are important reasons for considering both. Along passive margins with broad shelves like the Atlantic coast, the continental shelf is the relevant spatial unit for discussing carbon fluxes. Along

active margins with narrow shelves, such as the Pacific coast, a larger region than just the shelf needs to be considered to meaningfully discuss coastal carbon dynamics. The 370-km limit chosen here to define coastal oceans was recommended by Hales et al. (2008) and corresponds to the Exclusive Economic Zone (EEZ, the region where a nation can claim exclusive rights for fishing, drilling, and other economic activities). Worth noting here is that ocean $CO_2$ uptake or loss is not credited to any nation under Intergovernmental Panel on Climate Change (IPCC) $CO_2$ accounting; instead, ocean uptake is viewed as an

internationally shared public commons.

This review is structured as follows. First, we summarize the key variables and fluxes relevant to carbon budgets for coastal waters and describe the mechanisms by which carbon can be removed from the atmospheric reservoir and the means for quantifying the resulting carbon removal (see Section 2). Next we present available research relevant to carbon budgets for North American coastal waters by region, along with an assessment of whether enough information is available to derive

robust estimates of carbon export to the open ocean (see Section 3). Last, we discuss climate-driven trends in coastal carbon fluxes and coastal ocean acidification (see Section 4), followed by conclusions.

## 2 General overview of coastal carbon fluxes and stocks

Carbon is present in various inorganic and organic forms in coastal waters (Figure 2). Dissolved inorganic species include aqueous $CO_2$ (a combination of dissolved $CO_2$ and carbonic acid), bicarbonate and carbonate ions, and methane ($CH_4$); the

first three carbon species are collectively referred to as dissolved inorganic carbon or DIC. The major particulate inorganic species is calcium carbonate ($CaCO_3$), also referred to as particulate inorganic carbon (PIC). In addition to its inorganic forms, carbon is present in various dissolved and particulate organic forms (DOC and POC). In shelf waters, the reduced carbon

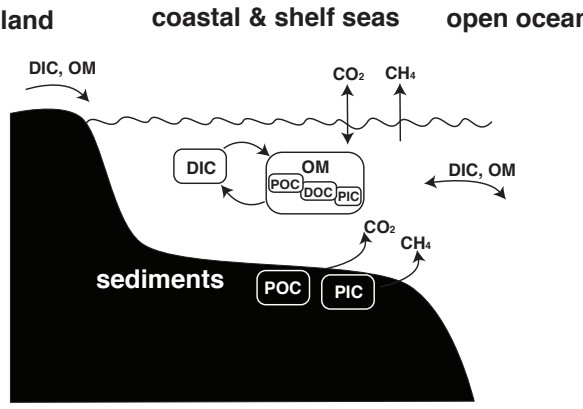

**Figure 2.** Major carbon pools and fluxes in coastal waters. Carbon pools include dissolved inorganic carbon (DIC), organic matter (OM), dissolved organic carbon (DOC), particulate organic carbon (POC), and particulate inorganic matter (PIC).

pool or total organic carbon pool (TOC) represents roughly 2% to 5% of the total carbon stock (Liu et al., 2010), and DOC constitutes more than 90% to 95% of this TOC (Vlahos et al., 2002).

Carbon is constantly transferred among these different pools and exchanged across the interfaces that demarcate coastal waters: the land-ocean interface, the air-sea interface, and the interface between coastal and open-ocean waters (Figure 2). The

internal carbon transformations include photosynthetic primary production, respiration, transfers between lower and higher trophic levels of the food web, exchanges between sediment and overlying water, biogeochemical processes in the sediment, and the formation and dissolution of $CaCO_3$. Of major importance are the conversion of DIC into organic carbon (POC and DOC), through primary production, and the reverse transformation by respiration throughout the water column, returning most of the organic carbon back into DIC. Some POC settles out of the water column and becomes incorporated into the sediments

where most of this material is respired through a range of different redox processes that produce DIC and, in the relative absence of electron acceptors other than $CO_2$, $CH_4$. Both DIC and $CH_4$ are released back into the overlying water. POC that is not respired (referred to as refractory POC) can be buried in sediments and stored for a very long time. In addition to the formation of organic matter through photosynthesis, some organisms precipitate internal or external body structures of $CaCO_3$ (or PIC), which either dissolve or become incorporated into the sediments and are buried. This discussion will refer to

long-term storage of buried POC and PIC in coastal sediments as permanent burial.

A major carbon exchange process along ocean margins is the flux of $CO_2$ across the air-sea interface. The annual cycle of this flux is driven by the under- or oversaturation of surface ocean $CO_2$ resulting from ocean warming and cooling, which affects $CO_2$ solubility, from primary production, respiration, and $CaCO_3$ precipitation and dissolution, and from transport of DIC to and from the ocean surface (e.g., by upwelling and convection). Other factors that influence the resistance to gas

exchange across the air-sea interface are winds, sea ice extent, and surface films. The annual cycles of primary production, respiration, and air-sea $CO_2$ flux tend to be of larger magnitude and more variable in coastal waters than in the open ocean



(Bauer et al., 2013; Liu et al., 2010; Muller-Karger et al., 2005; Thunell et al., 2007). Other important exchange fluxes are organic and inorganic carbon inputs from land via rivers and estuaries (Najjar et al., 2018), from tidal wetlands (Ho et al., 2017), and exchanges between the coastal and open oceans across the continental shelf break or the operationally defined open-ocean boundary of the coastal ocean (Fennel, 2010). Net removal of carbon from direct interaction with the atmospheric

reservoir can occur by export of organic and inorganic carbon to the deep ocean or by permanent burial in coastal sediments.

Although continental shelves make up only 7% to 10% of the global ocean surface area, they are estimated to contribute up to 30% of primary production, 30% to 50% of inorganic carbon burial, and 80% of organic carbon burial (Dunne et al., 2007; Gattuso et al., 1998). As such, continental shelves have been argued to contribute disproportionately to the oceanic uptake of $CO_2$ (Cai, 2011; Liu et al., 2010; Muller-Karger et al., 2005).

Carbon export, referring to the flux of organic and inorganic carbon from coastal waters to the deep ocean, can occur through the so-called Continental Shelf Pump—a term coined by Tsunogai et al. (1999) after they observed a large uptake of atmospheric $CO_2$ in the East China Sea. There are two distinct mechanisms underlying the Continental Shelf Pump (Fennel, 2010). The first is physical in nature and thought to operate in mid- and high-latitude systems. In winter, shelf water is cooled more strongly than surface water in the adjacent open ocean because the former is not subject to deep convection. The colder

shelf water is denser and experiences a larger influx of atmospheric $CO_2$; both density and the solubility of $CO_2$ increase with decreasing temperature. If this dense and carbon-rich water is transported off the shelf, it will sink due to its higher density, and the associated carbon will be exported to the deep ocean. The second mechanism relies on biological processes that concentrate carbon below the seasonal pycnocline by photosynthetic production of organic carbon and subsequent sinking. If the carbon-rich water below the seasonal pycnocline is moved off the shelf horizontally, carbon potentially could be exported if this

water is transported or mixed below the seasonal thermocline. The depth to which the shelf-derived carbon can be exported is different for POC, which would sink, and DOC and DIC, which primarily would be advected laterally. Both mechanisms for carbon export critically depend on physical transport of carbon-rich water off the shelf.

Carbon export flux from coastal waters to the deep ocean cannot be quantified easily or accurately through direct observation. Thus, the only available estimates of such export are indirect, using mass balances of POC and dissolved oxygen (Hales et al.,

2006), mass balances of DOC (Barrón and Duarte, 2015; Vlahos et al., 2002), mass balances of TOC and DIC (Najjar et al., 2018), or model estimates (Izett and Fennel, 2018a, b; Bourgeois et al., 2016; Fennel and Wilkin, 2009; Mannino et al., 2016; Xue et al., 2013). If the total carbon inventory in a coastal system can be considered constant over a sufficiently long timescale (i.e., on the order of years), inferring carbon export is possible from using the sum of all other exchange fluxes across the system's interfaces over that same period. Export to the open ocean must balance the influx of carbon from land

and wetlands, its net exchange across the air-sea interface, lateral exchange caused by advection, and any removal through permanent sediment burial. The accuracy of the inferred export flux directly depends on the accuracy of the other flux estimates and of the assumption of a constant carbon inventory. Quantifying internal transformation processes (e.g., respiration, primary and secondary production) does not directly enter this budgeting approach but can elucidate the processes that drive fluxes across interfaces.



Current estimates of carbon fluxes across coastal interfaces come with significant uncertainties (Regnier et al., 2013). These uncertainties are caused by a combination of small-scale temporal and spatial variability, which is undersampled by currently available means of direct observation, and regional heterogeneity, which makes scaling up observations from one region to larger areas difficult. Contributing to variability in regional carbon budgets and export are geographical differences arising

from variations in shelf width, upwelling strength, the presence or absence of large rivers, seasonal ice cover, and latitude. Section 3 describes the regional characteristics of North American coastal waters and how these characteristics influence carbon dynamics. Available estimates of carbon fluxes are compiled in an attempt to estimate export.

The motivation for quantifying permanent burial of carbon and export of carbon from coastal waters to the deep ocean is that both processes remove $CO_2$ from the atmospheric reservoir. A more relevant but harder to obtain quantity in this context is

the burial or export of anthropogenic carbon. The anthropogenic component of a given carbon flux is defined as the difference between its preindustrial and present-day fluxes. Thus, present-day carbon fluxes represent a superposition of the anthropogenic flux component and the natural background flux. Only total fluxes—the sum of anthropogenic and background fluxes—can be observed directly. Distinction between anthropogenic fluxes and the natural background is difficult to assess for coastal ocean fluxes and has to rely on process-based arguments and models (Regnier et al., 2013) .

## 15  3  Review of coastal carbon fluxes around North America

In this section we briefly describe the bathymetric and hydrographical features of the four major North American coastal margins: the Atlantic coast, the Pacific coast, the coast of the northern Gulf of Mexico, and the Arctic coast, followed by a review of available carbon flux estimates for each. Where multiple flux estimates are available for the same region it is important to keep in mind that their spatial footprints and time windows do not necessarily match exactly.

### 20  3.1  Atlantic coast

The North American Atlantic coast borders on a wide, geologically passive-margin shelf that extends from the southern tip of Florida to the continental shelf of the Labrador Sea (Figure 1). The shelf is several hundred kilometers wide in the north (Labrador shelf and Grand Banks) but narrows progressively toward the south in the Mid Atlantic Bight (MAB), which is between Cape Cod and Cape Hatteras, and the South Atlantic Bight (SAB), which is south of Cape Hatteras. The SAB shelf

width measures only several tens of kilometers.

Two major semi-enclosed bodies of water are the Gulf of Maine (GOM) and Gulf of St. Lawrence. Important rivers and estuaries north of Cape Hatteras include the St. Lawrence River and Estuary, the Hudson River, Long Island Sound, Delaware Bay, and Chesapeake Bay. South of Cape Hatteras, the coastline is characterized by small rivers and marshes.

The SAB is impacted by the Gulf Stream, which flows northeastward along the shelf edge before detaching at Cape Hatteras

and meandering eastward into the open North Atlantic Ocean. North of Cape Hatteras, shelf circulation is influenced by the confluence of the southwestward-flowing fresh and cold shelf-break current (a limb of the Labrador Current) and the warm and salty Gulf Stream (Loder, 1998). Because shelf waters north of Cape Hatteras are sourced from the Labrador Sea, they are



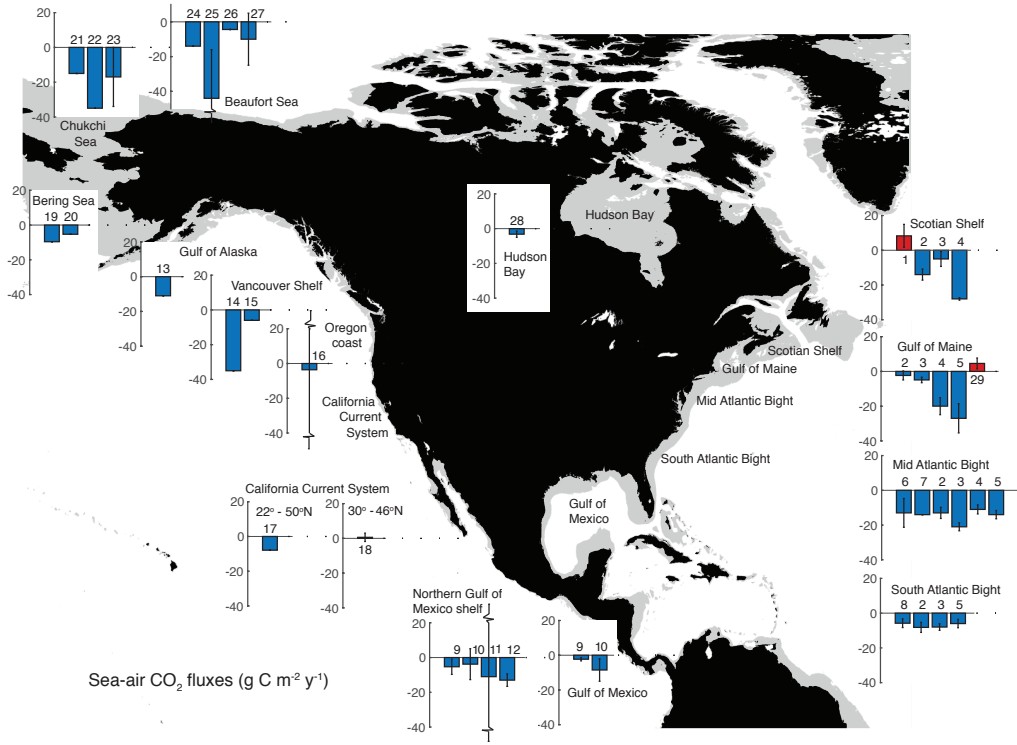

**Figure 3.** Observation- and model-based estimates of regional net air-sea $CO_2$ flux in g C m$^{-2}$ y$^{-1}$. Positive fluxes (red bars) indicate a flux to the atmosphere. Broken error bars are used where errors bars reach outside the range of the y-axis. References are: 1–observation-based estimate by Shadwick et al. (2010), 2–observation-based estimate by Signorini et al. (2013) using Ho et al. (2011) gas transfer parameterization, 3–satellite-based estimate from 2, 4–model estimate by Fennel and Wilkin (2009), 5–model-based estimate by Cahill et al. (2016), 6–observation-based estimate by DeGrandpre et al. (2002), 7–model-based estimate by Fennel et al. (2008), 8–observation-based estimate by Jiang et al. (2008), 9–observation-based estimate by Robbins et al. (2014), 10–model-based estimate by Xue et al. (2016b), 11–observation-based estimate by Huang et al. (2015), 12–satellite-based estimate by Lohrenz et al. (2018), 13–observation-based estimate by Evans and Mathis (2013), 14–observation-based estimate by Evans et al. (2012), 15–model-based estimate by Ianson and Allen (2002), 16–observation-based estimate by Evans et al. (2011), 17–satellite-based estimate by Hales et al. (2012), 18–model-based estimate by Turi et al. (2014), 19–observation-based estimate by Cross et al. (2014b), 20–observation-based estimate by Takahashi et al. (2009), 21–observation-based estimate by Evans et al. (2015b), 22–observation-based estimate by Gao et al. (2012), 23–observation-based estimate by Yasunaka et al. (2016), 24–observation-based estimate by Shadwick et al. (2011), 25–observation-based estimate by Else et al. (2013), 26–observation-based estimate by Evans et al. (2015b), 27–observation-based estimate by Mucci et al. (2010), 28–observation-based estimate by Else et al. (2008), 29–observation-based estimate by Vandemark et al. (2011). The flux estimates are also reported in Table S1.



relatively cold, fresh, and carbon rich, while slope waters (those located between the shelf break and the northern wall of the Gulf Stream) are a mixture of Labrador Current and Gulf Stream water. South of Cape Hatteras, exchange between the shelf and open ocean across the shelf break is impeded by the presence of the Gulf Stream and occurs via baroclinic instabilities in its northern wall (Lee et al., 1991). In the MAB and on the Scotian Shelf, cross-shelf exchange is hindered by shelf-break jets

and fronts (Rutherford and Fennel, 2018).

Air-sea fluxes of $CO_2$ exhibit a large-scale latitudinal gradient along the Atlantic coast (Figure 3) and significant seasonal and interannual variability (Figure 4). Discrepancies in independent estimates of net air-sea flux are largest for the Scotian Shelf and the Gulf of Maine (Figure 3). For the Scotian Shelf, Shadwick et al. (2010), combining in situ and satellite observations, reported a large source of $CO_2$ to the atmosphere of $8.3 \pm 6.6$ g C m$^{-2}$ y$^{-1}$. In contrast, Signorini et al. (2013) estimated a

relatively large sink of atmospheric $CO_2$, $14 \pm 3.2$ g C m$^{-2}$ y$^{-1}$, when using in situ data alone and a much smaller uptake, $5.0 \pm 4.3$ g C m$^{-2}$ y$^{-1}$, from a combination of in situ and satellite observations. The open GOM (excluding the tidally mixed Georges Bank and Nantucket Shoals) was reported as a weak net source of $4.6 \pm 3.1$ g C m$^{-2}$ y$^{-1}$ by Vandemark et al. (2011) but with significant interannual variability, while Signorini et al. (2013) estimate the region to be neutral (Table S1). The shallow, tidally mixed Georges Bank and Nantucket Shoals are thought to be sinks, however (Table S1).

The MAB and SAB are consistently estimated to be net sinks. Observation-based estimates for the MAB sink are $13 \pm 8.3$ g C m$^{-2}$ y$^{-1}$ (DeGrandpre et al., 2002) and $13 \pm 3.2$ g C m$^{-2}$ y$^{-1}$ (Signorini et al., 2013). Estimates for the SAB sink are $5.8 \pm 2.5$ g C m$^{-2}$ y$^{-1}$ (Jiang et al., 2008) and $8.2 \pm 2.9$ g C m$^{-2}$ y$^{-1}$ (Signorini et al., 2013). The transition from neutral or occasional net source on the Scotian Shelf and in the GOM to net sink in the MAB arises because the properties of shelf water are modified during its southwestward flow by air-sea exchange, inflows of riverine and estuarine waters (Salisbury

et al., 2008, 2009), and exchange with the open North Atlantic across the shelf break (Cai et al., 2010b; Wang et al., 2013). The cold, carbon-rich water on the Scotian Shelf carries a pronounced signature of its Labrador Sea origin. The GOM, which is deeper than the Scotian Shelf and the MAB and connected to the open North Atlantic through a relatively deep channel, is characterized by a mixture of cold, carbon-rich shelf waters entering from the Scotian Shelf and warmer, saltier slope waters. Shelf water in the MAB is sourced from the GOM and thus is a mixture of Scotian Shelf and slope water.

Shelf water in the SAB is distinct from that in the MAB and has almost no trace of Labrador Current water; instead, its characteristics are similar to those of the Gulf Stream, but its carbon signature is modified by significant organic and inorganic carbon and alkalinity inputs from coastal marshes (Cai et al., 2003; Jiang et al., 2013; Wang and Cai, 2004; Wang et al., 2005). Herrmann et al. (2015) estimated that 59% of the 3.4 Tg C y$^{-1}$ of organic carbon export from U.S. East Coast estuaries occurs in the SAB. The subsequent respiration of this organic matter and direct outgassing of marsh-derived carbon make the

nearshore regions a significant $CO_2$ source almost year-round. Despite the carbon inputs from marshes, uptake of $CO_2$ on the mid- and outer shelf regions during the winter months is large enough to balance $CO_2$ outgassing in the other seasons and on the inner shelf, making the SAB an overall weak sink (Jiang et al., 2008).

North of Cape Hatteras, $CO_2$ dynamics are characterized by strong seasonality with solubility-driven uptake by cooling in winter and biologically driven uptake in spring followed by outgassing in summer and fall due to warming and respiration of

organic matter (Figure 4d; DeGrandpre et al., 2002; Shadwick et al., 2010; Signorini et al., 2013; Vandemark et al., 2011; Wang





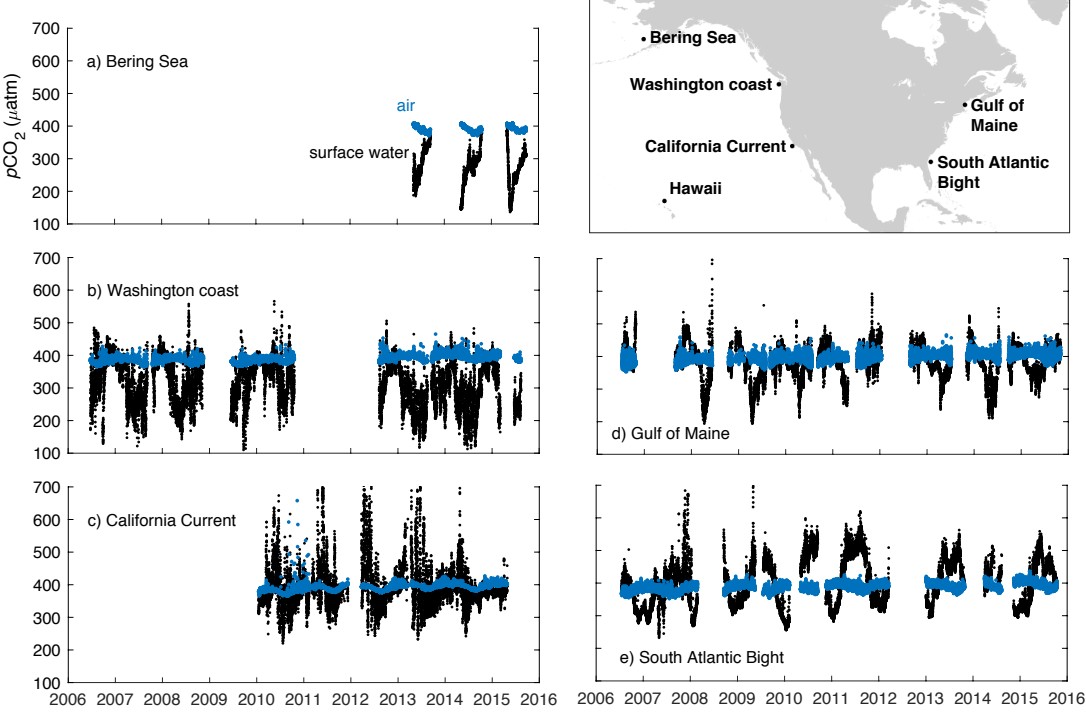

**Figure 4.** Observations of $p$CO$_2$ (in $\mu$atm) in the surface ocean (black) and overlying atmosphere (blue) at five coastal moorings. Map shows mooring locations. Also shown on the map is the location of the Hawaii Ocean Time Series (see Figure 6). Data sources: Bering Sea (mooring M2; Cross et al., 2014a); Washington coast (Cape Elizabeth mooring; Mathis et al., 2013); California Current (mooring CCE2; Sutton et al., 2012); Coastal Western Gulf of Maine mooring (Sutton et al., 2013); South Atlantic Bight (Gray's Reef mooring; Sutton et al., 2011)

et al., 2013). Hydrography and CO$_2$ dynamics on the Scotian Shelf are influenced by the significant freshwater input from the St. Lawrence River. Riverine inputs of carbon and nutrients are relatively small in the GOM but can cause local phytoplankton blooms, CO$_2$ drawdown, and low-pH conditions (Salisbury et al., 2008, 2009). Riverine and estuarine inputs become more important in the MAB with discharges from the Chesapeake Bay and the Delaware, Hudson, and Connecticut rivers (Wang

5   et al., 2013).

South of Cape Hatteras, seasonal phytoplankton blooms do not occur regularly and biologically driven CO$_2$ uptake is less pronounced than that further north (Figure 4e; Reimer et al., 2017; Wang et al., 2013), although sporadic phytoplankton blooms do occur because of intrusions of high-nutrient subsurface Gulf Stream water (Wang et al., 2005, 2013). The influence of riverine inputs is small and localized in the SAB (Cai and Wang, 1998; Wang and Cai, 2004; Wang et al., 2005; Xue et al.,

10   2016a; Reimer et al., 2017).

Regional biogeochemical models reproduce the large-scale patterns of air-sea CO$_2$ flux with oceanic uptake increasing from the SAB to the GOM (Cahill et al., 2016; Fennel et al., 2008; Previdi et al., 2009). These model studies elucidate





the magnitude and sources of interannual variability as well as long-term trends in air-sea $CO_2$ fluxes. Previdi et al. (2009) investigated opposite phases of the North Atlantic Oscillation (NAO) and found that the simulated air-sea flux in the MAB and GOM was 25% lower in a high-NAO year compared to a low-NAO year. In the MAB, the decrease resulted primarily from changes in wind forcing, while in the GOM changes in surface temperature and new production were more important. Cahill

et al. (2016) investigated the impact of future climate-driven warming and trends in atmospheric forcing (primarily wind) on air-sea $CO_2$ flux (without considering the atmospheric increase in $CO_2$). Their results suggest that warming and changes in atmospheric forcing have modest impacts on air-sea $CO_2$ flux in the MAB and GOM compared to that in the SAB where surface warming would turn the region from a net sink into a net source of $CO_2$ to the atmosphere if the increase of atmospheric $CO_2$ were ignored. Model studies also illustrate the effects of interactions between biogeochemical transformations in the sediment

and the overlying water column on carbon fluxes. For example, Fennel et al. (2008) showed that the effective alkalinity flux resulting from denitrification in sediments of the Atlantic coast reduces the simulated ocean uptake of $CO_2$ by 6% compared to a simulation without sediment denitrification.

The passive-margin sediments along the Atlantic coast have not been considered an area of significant $CH_4$ release until recently (Brothers et al., 2013; Phrampus and Hornbach, 2012; Skarke et al., 2014). Phrampus and Hornbach (2012) predicted

that massive seepage of $CH_4$ from upper-slope sediments is occurring in response to warming of intermediate-depth Gulf Stream waters. Brothers et al. (2013) and Skarke et al. (2014) documented widespread $CH_4$ plumes in the water column and attributed them to gas hydrate degradation. Estimated $CH_4$ efflux from the sediment in this region ranges from $1.5 \times 10^{-5}$ to $1.8 \times 10^{-4}$ Tg y$^{-1}$, where the uncertainty range reflects different assumptions underlying the conversion from $CH_4$ plume observations to seepage rates. The fraction of the released $CH_4$ that escapes to the atmosphere remains uncertain (Phrampus

and Hornbach, 2012).

### 3.2 Pacific coast

The North American Pacific coast extends from Panama to the Gulf of Alaska and is an active margin with varying shelf widths (Figure 1). The continental shelf is narrow along the coasts of California, Oregon, and Washington, on the order of 10 km, but widening significantly in the Gulf of Alaska, where shelves extend up to 200 km offshore. In the Gulf of Alaska,

freshwater and tidal influences strongly affect cross-shelf exchange, and the shelf is dominated by a downwelling circulation. The region from Vancouver Island to Baja California is a classic eastern boundary current upwelling region—the California Current System (Chavez et al., 2017; Hickey, 1998). Winds drive a coastal upwelling circulation characterized by equatorward flow in the California Current and by coastal jets and their associated eddies and fronts that extend offshore, particularly off the coasts of Baja California, California, Washington, and Oregon (Huyer, 1983).

The northern California Current System experiences strong freshwater influences and seasonality in wind forcing that diminish toward the south. In addition to the Columbia River and the Fraser River, a variety of small mountainous rivers, with highly variable discharge, supply freshwater. The Central American Isthmus runs from Panama to the southern tip of Baja California and experiences intense and persistent wind events, large eddies, and high waves that combine to produce upwelling and strong nearshore mixing (Chapa-Balcorta et al., 2015; Franco et al., 2014). In addition to alongshore winds, strong seasonal wind



jets that pass through the Central American cordillera create upwelling hotspots and drive production during boreal winter months in the gulfs of Tehuantepec, Papagayo, and Panama (Brown et al., 2015; Chapa-Balcorta et al., 2015; Chelton et al., 2000a, b; Gaxiola-Castro and Muller-Karger, 1998). The California Current brings water from the North Pacific southward into the southern California and Central American Isthmus regions, while the California Undercurrent transports equatorial waters

northward in the subsurface (Hickey, 1998).

The net exchange of $CO_2$ with the atmosphere along the Pacific coast is characterized by strong spatial and temporal variation and reflects complex interactions between biological uptake of nutrients and degassing of nutrient- and carbon-rich upwelled waters (Figure 4b, c). A growing number of coastal air-sea flux studies have used extrapolation techniques to estimate fluxes across the coastal oceans on regional to continental scales (Figure 3). Observation-based studies of air-sea $CO_2$ flux suggest

that estimates for the coastal ocean from Baja California to the Gulf of Alaska range from a weak (Coronado-Álvarez et al., 2017) to moderate sink of atmospheric $CO_2$ over this broad longitudinal range. Central California coastal waters have long been understood to have near-neutral air-sea $CO_2$ exchange because of their large and counter-balancing periods of efflux during upwelling conditions and influx during periods of relaxation and high primary productivity; this pattern is strongly modulated by El Niño–La Niña conditions (Friederich et al., 2002).

Hales et al. (2005) used seasonal data to estimate an uptake of 88 g C m$^{-2}$ y$^{-1}$ by Oregon coastal waters, which is about 15 times larger than the global mean of 6 g C m$^{-2}$ y$^{-1}$. In a follow-up analysis with greater temporal coverage, Evans et al. (2011) showed how large flux events can significantly alter the estimation of net exchanges for the Oregon shelf. After capturing a large and short-lived efflux event, their annual estimate was outgassing of 3.1 ± 82 g C m$^{-2}$ y$^{-1}$ for this same region. The disparity illustrates the importance of basing regional flux estimates on observations that are well resolved in time and space.

Capitalizing on the increased and more uniform spatiotemporal coverage of satellite data, Hales et al. (2012) estimated an annual mean uptake of 7.9 g C m$^{-2}$ y$^{-1}$ between 22$^o$ and 50$^o$N within 370 km offshore. The most northern estimates for the Pacific coast by Evans et al. (2012) and Evans and Mathis (2013) are influxes of 26 g C m$^{-2}$ y$^{-1}$ for British Columbian coastal waters shoreward of the 500-m isobath and 18 g C m$^{-2}$ y$^{-1}$ for Gulf of Alaska coastal waters shoreward of the 1500-m isobath.

Models for the upwelling region (Fiechter et al., 2014; Turi et al., 2014) reproduce the pattern of $CO_2$ outgassing nearshore and $CO_2$ uptake further offshore. They also illustrate the intense eddy-driven variability nearshore. Turi et al. (2014) simulate a weak source of 0.6 ± 2.4 g C m$^{-2}$ y$^{-1}$ for the region from 30$^o$ to 46$^o$N extending 800 km of shore. In contrast, Hales et al. (2012) reported a sink of 7.9 g C m$^{-2}$ y$^{-1}$, based on observations, for the same latitudinal band but only extending 370 km of shore. Fiechter et al. (2014) simulate a source of atmospheric $CO_2$ of 0.6 Tg C y$^{-1}$ for the region from 35$^o$ to

45$^o$N within 600 km of shore, also in contrast to the observation-based estimate of a 14 Tg C sink published by Hales et al. (2012). (The estimate of Fiechter et al. (2014) is not inlcuded in Figure 3 because the area-normalized flux is not available from that study.) Both models simulate strong outgassing within the first 100 km of shore, driven by intense upwelling of nutrient- and carbon-rich water, compensated by biologically driven $CO_2$ uptake from the atmosphere as upwelled nutrients are consumed by photosynthesis during subsequent offshore advection within several hundreds of kilometers of the coast. The

disagreement in mean simulated fluxes between the two models may result partly from different choices of averaging region





and period and differences in model forcing, such as the climatological forcing in Turi et al. (2014) versus realistic variability in Fiechter et al. (2014). Notable, observations for the Oregon shelf by Evans et al. (2015a) showed intense summer upwelling that led to strong outgassing with pronounced variability in air-sea fluxes but only weak stimulation of primary production. They hypothesized that nutrient-rich waters might be subducted offshore at convergent surface temperature fronts before nutrients are fully consumed by primary producers.

Cross-shelf exchange of carbon occurs in the California Current System mostly in response to wind-driven circulation and eddies, but river plumes and tides also have been shown to increase offshore transport in the northern part of the system (Barth et al., 2002; Hales et al., 2006). Uncertainties in published estimates are high, ranging from very small (Ianson and Allen, 2002; Pennington et al., 2010) to very high fractions of primary production (Hales et al., 2005; Turi et al., 2014), again as a result of the region's large spatial and temporal variability.

Less is known about the air-sea flux of $CH_4$ along the Pacific margin. Recent studies inventoried sedimentary sources of $CH_4$ hydrates, derived from terrestrial and coastal primary production, and suggested that extensive deposits along the Cascadia margin are beginning to destabilize because of warming (Hautala et al., 2014; Johnson et al., 2015).

### 3.3 Gulf of Mexico

The Gulf of Mexico is a semi-enclosed marginal sea at the southern coast of the conterminous United States. The passive margin shelves of its northern portion are relatively wide (up to 250 km west of Florida) but, in contrast to shelf waters of the Atlantic coast, those of the Gulf of Mexico are not separated from open-ocean waters by shelf-break fronts or currents. Ocean water enters the Gulf mainly through the Yucatan Channel, where it forms the northeastward meandering Loop Current, which sheds anticyclonic eddies and exits the Gulf through the Florida Straits (Muller-Karger et al., 2015; Rivas et al., 2005). While shelf circulation is influenced primarily by local wind and buoyancy forcing, outer-shelf regions are at times influenced by Loop Current eddies that impinge on and interact with the shelf (Lohrenz and Verity, 2004). Riverine input is substantial in the northern Gulf of Mexico, where the Mississippi-Atchafalaya river system delivers large loads of freshwater, nutrients, and sediments.

Estimates of air-sea $CO_2$ flux are available from observations and model simulations (Figure 3). Observational estimates indicate that Gulf of Mexico, as a whole, is a weak net sink of atmospheric $CO_2$ with an annual average of $2.3 \pm 1.0$ g C m$^{-2}$ y$^{-1}$ (Robbins et al., 2014). Robbins et al. (2014) also provide flux estimates, as follows, for smaller shelf regions, namely, the West Florida Shelf, the northern Gulf shelf, the western Gulf shelf, and the Mexico shelf. The West Florida Shelf and western Gulf shelf act as sources to the atmosphere, with estimated annual average fluxes of $4.4 \pm 1.3$ and $2.2 \pm 0.6$ g C m$^{-2}$ y$^{-1}$, respectively. The northern Gulf acts as a sink, with an estimated flux of $5.3 \pm 4.4$ g C m$^{-2}$ y$^{-1}$, and the Mexican shelf is almost neutral, with an estimated uptake flux of $1.1 \pm 0.6$ g C m$^{-2}$ y$^{-1}$. A more recent estimate for the West Florida Shelf is $4.4 \pm 18.0$ g C m$^{-2}$ y$^{-1}$ (Robbins et al., 2018). Huang et al. (2015) estimated a larger uptake on the northern Gulf shelf of $11 \pm 44$ g C m$^{-2}$ y$^{-1}$ (i.e., about twice the estimate of Robbins et al. (2014)) and reported a much larger uncertainty. In an analysis that combines satellite and in situ observations, Lohrenz et al. (2018) estimated a similar uptake for the northern Gulf





of Mexico of $13 \pm 3.6$ g C m$^{-2}$ y$^{-1}$. The overall carbon exchanges in the Gulf vary significantly from year to year because of interannual variability in wind, temperature, and precipitation (Muller-Karger et al., 2015).

Model-simulated air-sea CO$_2$ fluxes by Xue et al. (2016b) agree relatively well with the estimates of Robbins et al. (2014), reproducing the same spatial pattern though their simulated Gulf-wide uptake of $8.5 \pm 6.5$ g C m$^{-2}$ y$^{-1}$ is larger. This

discrepancy results largely from a greater simulated sink in the open Gulf. Also, the uncertainty estimates of the model-simulated fluxes by Xue et al. (2016b) are much larger than those of Robbins et al. (2014); the latter might be too optimistic in reporting uncertainties of the flux estimates. Overall, the various observation- and model-derived estimates for Gulf regions agree in terms of their broad patterns, but existing discrepancies and, at times, large uncertainties indicate that current estimates need further refinement.

Quantitative understanding of CH$_4$ dynamics in coastal and oceanic environments of the Gulf of Mexico is limited. Solomon et al. (2009) speculated that deep CH$_4$ hydrate seeps in the Gulf potentially are a significant CH$_4$ source to the atmosphere. They estimated ocean-atmosphere fluxes from seep plumes of $1{,}150 \pm 790$ to $38{,}000 \pm 21{,}000$ g CH$_4$ m$^{-2}$ d$^{-1}$ compared with $2.2 \pm 2.0$ to $41 \pm 8.2$ g CH$_4$ m$^{-2}$ d$^{-1}$ for background sites. Subsequent acoustic analyses of bubble plume characteristics question the finding that CH$_4$ bubbles make their way to the surface (Weber et al., 2014), and the fate of CH$_4$ emissions from

seeps and their overall contribution to atmospheric CH$_4$ remain uncertain.

### 3.4  Arctic coast

The North American Arctic coastal ocean comprises broad ($\sim$300 km) shallow shelves in the Bering and Chukchi seas, the narrower (<100-km) Beaufort Sea shelf, Hudson Bay, and the extensive Canadian Arctic shelf (Figure 1). Shelf water enters these regions from the North Pacific and follows a large-scale pathway from its entrance through the Bering Strait via the

Chukchi and Beaufort seas onto the Canadian Arctic shelf and, ultimately, the North Atlantic (Carmack et al., 2006, 2015). Hudson Bay receives significant inputs of freshwater (Déry et al., 2005). Except for the southernmost Bering Sea, most of the coastal Arctic is covered with sea ice from about October to June. Areas of persistent multiyear sea ice at the northernmost extent of the Canadian Arctic shelf are rapidly declining (Stroeve et al., 2012).

Globally, the pace of increasing air temperatures is highest in the Arctic, resulting in significant reductions in both summer

and winter sea ice cover that profoundly affect marine ecosystems (Moore and Stabeno, 2015; Steiner et al., 2015). The Arctic coast is sparsely populated with communities heavily reliant on subsistence fishing and hunting, and thus strongly affected by the rapid changes associated with global warming.

Coastal waters in the Arctic have been described consistently as a net sink for atmospheric CO$_2$ (Figure 3; Bates, 2006; Bates et al., 2011; Cross et al., 2014b; Else et al., 2008, 2013; Evans et al., 2015b; Gao et al., 2012; Mucci et al., 2010; Semiletov et al.,

2007; Shadwick et al., 2011). This general trait is caused by low surface water $p$CO$_2$ relative to the atmosphere during ice-free months (see, e.g., Figure. 4a). These low levels are set by a combination of low water temperatures and seasonally high rates of both ice-associated and open-water primary production (Cai et al., 2010a, 2014; Steiner et al., 2014), as well as by limited gas exchange through sea ice relative to open water (Butterworth and Miller, 2016; van der Loeff et al., 2014) during winter months when under-ice $p$CO$_2$ is higher. The degree to which gas exchange through sea ice is suppressed is debated within the





Arctic $CO_2$ flux community, partly as a result of inconsistencies between measurement methodologies and challenges of data collection in such a harsh environment, particularly during winter. The typical approach of calculating air-sea $CO_2$ flux (from measured air-sea $pCO_2$ differences and gas transfer rates parameterized using wind speed relationships) can differ markedly from flux estimations determined by eddy correlations. The latter suggest high rates of $CO_2$ exchange relative to open water
(Else et al., 2011). Three arguments indicate that the high, eddy correlation-based fluxes may be overestimates: the potential for unaccounted $CO_2$ and water vapor cross-correlation possibly affecting the measurement (Landwehr et al., 2014); independent analysis of the $^{222}$Radon isotope showing near-zero gas exchange in areas covered by sea ice (van der Loeff et al., 2014); and recent demonstration of dampened gas-transfer velocities via concurrent, properly corrected eddy covariance-based fluxes and air-sea $pCO_2$ difference measurements in the Antarctic marginal ice zone supporting linear scaling methods that calculate
fluxes using percent sea ice cover (Butterworth and Miller, 2016).

Despite the dampening effect of sea ice, its permeability is known to be a function of temperature (Golden et al., 2007). Therefore, as Arctic winter temperatures continue to rise, the role of wintertime air-ice $CO_2$ exchange may become increasingly important because rising temperatures may allow some degree of exchange to take place. To date, measurements of wintertime exchange have been limited to very few studies (Else et al., 2011, 2013; Miller et al., 2015). In recent years, the role of sea
ice growth and decay has been shown to significantly affect the air-sea $CO_2$ flux (Rysgaard et al., 2007, 2009). During sea ice formation, brine rejection forms dense high-saline water that is exported from the surface layer. This process alters the ratio of total alkalinity to sea ice DIC and the underlying seawater, because DIC is a component of the brine whereas total alkalinity precipitates in the brine channels as a form of $CaCO_3$ known as ikaite (Dieckmann et al., 2008; Rysgaard et al., 2013). During sea ice decay, ikaite dissolves, leading to excess total alkalinity relative to DIC and undersaturation of $CO_2$ in meltwater.

With regard to Arctic $CH_4$ fluxes, much more is known about the emission potential, distribution, and functioning of terrestrial sources (McGuire et al., 2009); knowledge of marine $CH_4$ sources is developing slowly due to sparse observations and the logistical challenges of Arctic marine research. The largest marine $CH_4$ source in the Arctic is dissociation of gas hydrates stored in continental margin sediments (Parmentier et al., 2013). As sea ice cover continues to retreat and ocean waters warm, $CH_4$ hydrate stability is expected to decrease with potentially large and long-term implications. An additional potential marine
$CH_4$ source, unique to polar settings, is release from subsea permafrost layers, with fluxes from thawed sediments reported to be orders of magnitude higher than fluxes from adjacent frozen sediments (Shakhova et al., 2015).

### 3.5 Summary estimates of $CO_2$ uptake and a carbon budget for North American coastal waters

Despite the variability in regional estimates discussed above and summarized in Figure 3 and Table S1, North American coastal waters clearly act as a net sink of atmospheric carbon. Because of discrepancies among studies and gaps in spatial coverage
it would be difficult to combine these various regional estimates into one number for North American coastal waters with any confidence. Instead, we consider estimates of net air-sea $CO_2$ exchange from two global data syntheses (Chen et al., 2013; Laruelle et al., 2014) and a process-based global model (Bourgeois et al., 2016). The data syntheses use a global segmentation of the coastal zone and associated watersheds known as MARCATS (MARgins and CATchments Segmentation; Laruelle et al., 2013), which, at a resolution of $0.5^o$, delineates a total of 45 coastal segments, eight of which surround North America. The





**Table 1.** Regional estimates of net air-sea $CO_2$ flux from two data syntheses and a global biogeochemical model for the MARgins and CATchments Segmentation (MARCATS) regions. Positive numbers indicate a flux to the atmosphere. 1 Tg = $10^{12}$ g

| MARCATS segment #, name, area ($10^3$ km$^2$) | Chen et al. (2013) | | Laruelle et al. (2014) | | Bourgeois et al. (2016) | |
|---|---|---|---|---|---|---|
| | mol C m$^{-2}$ y$^{-1}$ | Tg C y$^{-1}$ | mol C m$^{-2}$ y$^{-1}$ | Tg C y$^{-1}$ | mol C m$^{-2}$ y$^{-1}$ | Tg C y$^{-1}$ |
| 1, Northeastern Pacific, 460 | -41 | -19 | -15 | -6.8 | -22 | -10 |
| 2, California Current, 210 | -27 | -5.7 | -0.62 | -0.13 | -2.3 | -0.48 |
| 3, Tropical Eastern Pacific, 200 | -0.50 | -0.1 | 0.95 | 0.19 | -1.1 | -0.22 |
| 9, Gulf of Mexico, 540 | -2.4 | -1.3 | -3.9 | -2.1 | -8.3 | -4.5 |
| 10, Florida Upwelling, 860 | -13 | -11 | -3.1 | -2.7 | -17 | -15 |
| 11, Labrador Sea, 400 | -25 | -10 | -47 | -19 | -22 | -8.8 |
| 12, Hudson Bay, 1100 | 10 | 11 | NA | NA | -3.4 | -3.8 |
| 13, Canadian Arctic Archip., 1200 | -47 | -57 | -12 | -14 | -5.2 | -6.2 |
| Total, 4900 | | -94 | | -44 | | -49 |

data synthesis of Chen et al. (2013) is a summary of individual studies, whereas Laruelle et al. (2014) analyze the Surface Ocean $CO_2$ Atlas 2.0 database (Bakker et al., 2014) to derive regional estimates. The data syntheses of Chen et al. (2013) and Laruelle et al. (2014) estimate the North American coastal uptake to be 94.4 and 44.5 Tg C y$^{-1}$, respectively, and the process-based model of Bourgeois et al. (2016) estimates an uptake of 48.8 Tg C y$^{-1}$ (Table 1). Although there are significant

regional discrepancies between the latter two estimates for the eastern tropical Pacific Ocean (referred to here as the Central American Isthmus), the Gulf of Mexico, the Florida Upwelling region (actually covering the eastern United States including the SAB, MAB and GOM), the Labrador Sea, and the Canadian Arctic shelf, the total flux estimates for North America are in close agreement. This, and the fact that Laruelle et al. (2014) used an internally consistent methodology to estimate air-sea $CO_2$ fluxes, builds some confidence in these numbers.

The net $CO_2$ flux and its anthropogenic component from the process-based global model of Bourgeois et al. (2016) are also reported for a regional decomposition of the EEZs of the United States, Canada, and Mexico in Table 2. The model simulates a net uptake of $CO_2$ in North American EEZ coastal waters (excluding the EEZ of the Hawaiian and other islands) of 160 Tg C y$^{-1}$ with an anthropogenic flux contribution of 59 Tg C y$^{-1}$. We assume 160 Tg C y$^{-1}$ as the best estimate of net uptake by coastal waters of North America, excluding tidal wetlands and estuaries. Unfortunately, there are no formal

error estimates for this uptake. Instead, we estimate an error by noting first that the Bourgeois et al. (2016) model is in good agreement with the more recent of the two observation-based estimates for the MARCATS regions of North America, and furthermore, that the error estimate for the uptake by continental shelves globally is about 25%, with the North American





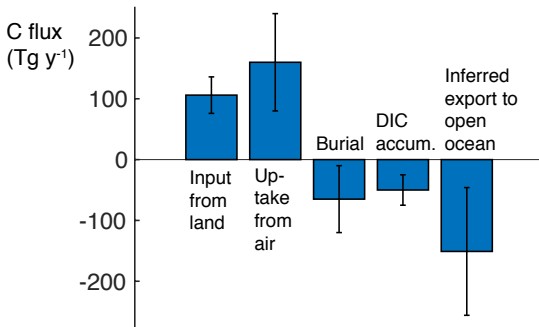

**Figure 5.** Carbon budget for the EEZ of the USA, Canada and Mexico excluding the EEZs of Hawaii and other islands. Here positive fluxes are a source to the coastal ocean. The accumulation of DIC in EEZ waters is reported with a negative sign to illustrate that all fluxes balance.

MARCATS regions having mainly "fair" data quality (Laruelle et al., 2014). Hence, assuming an error of ±50% for the uptake by North American EEZ waters seems reasonable.

Combining the atmospheric $CO_2$ uptake estimate with estimates of carbon transport from land and carbon burial in ocean sediments enables a first attempt at constructing a carbon budget for the North American EEZ. Carbon delivery to the coastal ocean from land via rivers and from tidal wetlands after estuarine processing (i.e., $CO_2$ outgassing and carbon burial in estuaries) is estimated to be $106 \pm 30$ Tg C y$^{-1}$ (Windham-Myers et al., 2018). Estimates of carbon burial, based on the method of Dunne et al. (2007) for the regional decomposition of the North American EEZ, are reported in Table 2, with a total flux of 120 Tg C y$^{-1}$. We consider these fluxes to be an upper bound because they are substantially larger than other estimates. The Dunne et al. (2007) global estimates of organic carbon burial in waters shallower than 2,000 m are $19\pm9$ g C m$^{-2}$ y$^{-1}$, much larger than the estimates of 6 and 1 g C m$^{-2}$ y$^{-1}$ by Chen (2004) and Muller-Karger et al. (2005), respectively, although areas are slightly different in the three studies. The organic carbon burial estimates of Dunne et al. (2007) for the GOM, MAB, and SAB (Table 2) are larger by factors of 8, 17, and 3, respectively, than the best estimates of the empirical model of (Najjar et al., 2018). However, due to different definitions of the boundary between coastal waters and the open ocean, the combined area of the GOM, MAB, and SAB in Najjar et al. (2018) is about a third of that of Dunne et al. (2007). Finally, Dunne et al. (2007) estimated the organic carbon burial in Hudson Bay to be 19 g C m$^{-2}$ y$^{-1}$, compared to a mean estimate of $1.5\pm0.7$ g C m$^{-2}$ y$^{-1}$ of burial from sediment cores (Kuzyk et al., 2009). Given these results, we consider the estimates of Dunne et al. (2007) to be an upper bound and assume that a reasonable lower bound is about an order of magnitude smaller, thus placing the organic carbon burial estimate at $65\pm55$ Tg C y$^{-1}$.

If these estimates of net air-sea flux, carbon burial, and carbon input from land are accurate, then the residual must be balanced by an increase in carbon inventory in coastal waters and a net transfer of carbon from coastal to open-ocean waters (Figure 5). In their global compilation, Regnier et al. (2013) report an increase in the coastal carbon inventory of 50 Tg C y$^{-1}$,

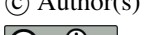


which is a quarter of their estimated anthropogenic carbon uptake by air-sea exchange in the coastal waters of 200 Tg C y$^{-1}$. The latter estimate is uncertain. In their global modeling study, which did not account for anthropogenic changes in carbon delivery from land, Bourgeois et al. (2016) estimated an accumulation of carbon in the coastal oceans of 30 Tg C y$^{-1}$. This amount is a third of their estimated uptake of anthropogenic carbon from air-sea gas exchange in the coastal ocean of

100 Tg C y$^{-1}$ and approximately half of their estimated cross-shelf export of anthropogenic carbon of 70 Tg C y$^{-1}$. The rate of carbon accumulation in the North American EEZ from the model of Bourgeois et al. (2016) is 50 Tg C y$^{-1}$ (Figure 5). Here again, we assume an uncertainty of ±50%. The residual of 151±105 Tg C y$^{-1}$ is the inferred export of carbon to the open ocean (Figure 5). The fact that the error in this residual is large in absolute and relative terms emphasizes the need for more accurate quantification of the terms in the coastal carbon budget. The challenge, however, is that many of these terms are

small compared to internal carbon cycling in coastal waters, which is dominated by primary production and decomposition. Two separate estimates of primary production (Table 2) are in broad agreement and reveal that the fluxes in Figure 5 (on the order of 10 to 100 Tg C y$^{-1}$) are just a few percent of primary production (on the order of 1000 Tg C y$^{-1}$, see Total in Table 2). This also indicates that small changes in carbon cycling in coastal waters can result in large changes in atmospheric uptake and transport to the open ocean.

## 4   Trends in carbon fluxes and acidification in North American coastal waters

### 4.1   Carbon flux trends

Important questions with respect to coastal carbon fluxes are: How will the coastal ocean change as a $CO_2$ sink? What is the anthropogenic component of the $CO_2$ sink? And how will changing climate and other forcings affect the total and anthropogenic flux proportions?

As stated in Section 2, when considering the ocean's role in sequestering anthropogenic carbon, the most relevant component is anthropogenic flux, not the total uptake flux. Neither quantifying the anthropogenic carbon flux component nor predicting its future trend is straightforward. Here the factors likely to result in trends in total carbon fluxes are described; by definition, changes in total carbon fluxes imply changes in anthropogenic fluxes as well.

A direct effect of increasing atmospheric $CO_2$ will be an increase in net uptake by the coastal ocean. In addition to rising

atmospheric $CO_2$ levels, changes in climate forcings (i.e., surface heat fluxes, winds, and freshwater input) and buffering capacity may affect carbon fluxes in coastal waters. Ocean warming reduces the solubility of gases and thus directly affects gas concentrations near the surface; this likely will decrease the net air-sea flux of $CO_2$ by reducing the undersaturation of $CO_2$ (see Cahill et al., 2016, for the North American Atlantic coast). Surface warming also strengthens vertical stratification and thus impedes vertical mixing, which will affect upward diffusion of nutrients and DIC. Enhanced stratification, therefore,

could lead to decreases in both biologically driven carbon uptake and $CO_2$ outgassing. However, model projections for the northern Gulf of Mexico show that the direct effect of increasing atmospheric $CO_2$ overwhelms the other more secondary effects (Laurent et al., 2018). Furthermore, temperature trends in coastal waters around North America show complex patterns with some regions having cooled from 1982 to 1997 followed by warming from 1997 to 2013 (e.g., the MAB), some regions



having warmed from 1982 to 1997 followed by cooling from 1997 to 2013 (e.g., the SAB and Gulf of Alaska), and other regions showing no consistent warming from 1982 to 2013 (e.g., the Arctic; Liao et al., 2015). Temperature anomalies from a time series in the central California Current System show warm surface waters for the decade prior to 1997 followed by a prolonged cooler period (Chavez et al., 2017). The cool period ended when strong surface warming, associated with a marine

heatwave and the 2015-2016 El Niño, interrupted the cool anomalies (Chavez et al., 2017). In the California Current System $pCO_2$ has increased faster than in the open ocean due to the combined effects of atmospheric uptake and an intensification of upwelling (Chavez et al., 2017). Deeper waters in the California Undercurrent have shown a multidecadal trend (1980 to 2012) toward warmer, saltier, lower-oxygen, and higher-$CO_2$ waters at a depth associated with increased northward transport of Pacific equatorial waters (Meinvielle and Johnson, 2013).

Some studies suggest that trends in the air-sea $pCO_2$ gradient ($\Delta pCO_2$) are indicative of a strengthening or weakening of the net $CO_2$ uptake by shelf systems, where an increasing $\Delta pCO_2$, implying that ocean $pCO_2$ rises more slowly than atmospheric $pCO_2$, corresponds to increased net uptake and cross-shelf export (Laruelle et al., 2018). In their observation-based analysis of decadal trends in shelf $pCO_2$, Laruelle et al. (2018) found that coastal waters lag compared to the rise in atmospheric $CO_2$ in most regions. They found that in the MAB $\Delta pCO_2$ has increased by $1.9 \pm 3.1$ $\mu$atm y$^{-1}$, a finding that suggests surface ocean

$pCO_2$ does not increase or increases at a rate that is substantially slower than in the atmosphere in this region. For the shelves of the Labrador Sea, Vancouver Shelf, and SAB, they found smaller rates of $0.68 \pm 0.61$ $\mu$atm y$^{-1}$, $0.83 \pm 1.7$ $\mu$atm y$^{-1}$, and $0.51 \pm 0.74$ $\mu$atm y$^{-1}$, respectively, again implying that surface ocean $pCO_2$ does not increase or increases at a slower rate than atmospheric $CO_2$. The only North American coastal region that exhibits a negative trend is the Bering Sea, with $-1.1 \pm 0.74$ $\mu$atm y$^{-1}$, meaning that surface ocean $pCO_2$ increases at a faster rate than in the atmosphere.

Laruelle et al. (2018) concluded that the lag in coastal ocean $pCO_2$ increase compared to that in the atmosphere in most regions indicates an enhancement in the coastal uptake and export of atmospheric $CO_2$, although they did not investigate alternative explanations. The observed lag may be influenced by decadal-scale climate variability. Earth system models suggest that it takes at minimum 30 years of observations in open-ocean regions to detect a change in the rate of anthropogenic carbon uptake above natural variability (McKinley et al., 2016). Assessments in coastal systems suggest the time to detect an

anthropogenic signal in seawater $pCO_2$ can be even longer there, making it even more difficult to directly observe anthropogenic changes in $CO_2$ uptake in highly variable coastal regions (Sutton et al., 2018).

Trends in coastal ocean uptake of $CO_2$ are highly variable regionally due to a complex interplay of factors. In coastal upwelling systems, surface warming will increase the horizontal gradient between cold, freshly upwelled source waters and warm, offshore surface water, leading to a greater tendency for the subduction of upwelled water at offshore surface temperature

fronts during periods of persistent and strong upwelling-favorable winds. The cumulative effect of these processes for the Pacific coast may be greater and more persistent $CO_2$ outgassing nearshore and lower productivity offshore as upwelled nitrate is exported before it can be used by the phytoplankton community (Evans et al., 2015a). Rates of warming clearly are faster in higher latitudes, but predicting the net effect of these warming-induced changes in the Arctic is not easy. Arctic warming also leads to reductions in ice cover, which directly affects air-sea gas exchange (Bates and Mathis, 2009). Another profound effect



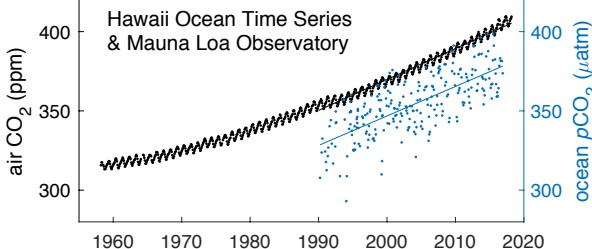

**Figure 6.** Atmospheric $CO_2$ (black dots) measured at the Mauna Loa Observatory in Hawaii beginning in 1958 and surface ocean $pCO_2$ data (blue dots) from the Hawaii Ocean Time Series (HOT) station near Hawaii (see Figure 4 for site location). Black and blue lines indicate linear trends after 1990. Atmospheric $CO_2$ increased by 1.86 ppm $y^{-1}$. Surface ocean $pCO_2$ data increased by 1.95 $\mu$atm $y^{-1}$. $pCO_2$ is calculated using CO2SYS with Mehrbach refit coefficients for the dissociation constants of $H_2CO_3$ and $HCO_3^-$, and Dickson's dissociation constant for $HSO_4^-$. Data sources: Mauna Loa, www.esrl.noaa.gov/gmd/ccgg/trends/data.html; HOT, hahana.soest.hawaii.edu/hot/hot-dogs/interface.html

of Arctic warming is the melting of permafrost, which leads to the release of large quantities of $CH_4$ to the atmosphere, from both the land surface and the coastal ocean (Crabeck et al., 2014; Parmentier et al., 2013).

Changes in wind stress also directly affect air-sea gas fluxes because stronger winds intensify gas exchange. For example, for the North American Atlantic coast, changes in wind stress were shown to significantly modify air-sea fluxes (Wanninkhof and

Trinanes, 2017; Cahill et al., 2016; Previdi et al., 2009). Large-scale changes in wind patterns also affect ocean circulation with a range of implications (Bakun, 1990). Upwelling-favorable winds along the North American Pacific coast have intensified in recent years, especially in the northern parts of the upwelling regimes (García-Reyes et al., 2015; Rykaczewski and Checkley, 2008; Rykaczewski et al., 2015; Sydeman et al., 2014). This has led to a shoaling of nutrient-rich subsurface waters (Aksnesa and Ohman, 2009; Bograd et al., 2015), increased productivity (Chavez et al., 2011, 2017; Jacox et al., 2015; Kahru et al.,

2015), higher DIC delivery to the surface (Turi et al., 2016), and declining oxygen levels (Crawford and Peña, 2016; Peterson et al., 2013; Bograd et al., 2015). In the coastal Arctic, late-season air-sea $CO_2$ fluxes may become increasingly directed toward the atmosphere as Arctic low-pressure systems with storm-force winds occur more often over open water, thus ventilating $CO_2$ respired from the high organic carbon loading of the shallow shelf (Evans et al., 2015b; Hauri et al., 2013; Steiner et al., 2013). The intense warming observed across the Arctic also influences mid-latitude weather patterns (Kim et al., 2014), likely with

cascading effects on $CO_2$ exchanges through adjustments in the wind field.

## 4.2 Acidification trends

Increasing atmospheric $CO_2$ emissions lead to rising atmospheric $CO_2$ levels (Figure 6) and a net ocean uptake of $CO_2$. Since about 1750, the ocean has absorbed 27% of anthropogenic $CO_2$ emissions to the atmosphere from fossil fuel burning, cement production, and land-use changes (Canadell et al., 2007; Le Quéré et al., 2015; Sabine and Tanhua, 2010). As a result of



this uptake, the surface ocean $p\text{CO}_2$ has increased (Figure 6) and oceanic pH, carbonate ion concentration, and carbonate saturation state have decreased (Caldeira and Wickett, 2003; Feely et al., 2004, 2009; Orr et al., 2005). Commonly called "ocean acidification," this suite of chemical changes is defined more precisely as "any reduction in the pH of the ocean over an extended period, typically decades or longer, which is caused primarily by uptake of $\text{CO}_2$ from the atmosphere but can also be caused by other chemical additions or subtractions from the ocean" (IPCC, 2011, p. 37). In addition to uptake of $\text{CO}_2$ from the atmosphere, variations in DIC concentrations and thus pH can be caused by biological production and respiration. Ocean

acidification can significantly affect growth, metabolism, and life cycles of marine organisms (Fabry et al., 2008; Gattuso and Hansson, 2011; Somero et al., 2015) and most directly affects marine calcifiers, organisms that precipitate $\text{CaCO}_3$ to form internal or external body structures. When the carbonate saturation state decreases below the equilibrium point for carbonate precipitation/dissolution, conditions are said to be corrosive, or damaging, to marine calcifiers. These conditions make it more difficult for calcifying organisms to form shells or skeletons, perform metabolic functions and survive. Early life stages are

particularly vulnerable as shown by recent large-scale die-offs of oyster larvae in the coastal Pacific where increased energetic expenses under low-pH have led to compromised development of essential functions and insufficient initial shell formation in oyster larvae (Waldbusser et al., 2015) as well as delayed and inhibited gametogenesis in oysters and sea urchins (Boulais et al., 2017).

Acidification trends in open-ocean surface waters tend to occur at a rate that is commensurate with the rate of the increase

in atmospheric $\text{CO}_2$ (see trends of atmospheric $\text{CO}_2$ in comparison to surface ocean $p\text{CO}_2$ at the Hawaii Ocean Time Series in Figure 6). Acidification in coastal waters is more variable (Sutton et al., 2016) because of a combination of changes in circulation and upwelling, larger amplitude seasonal signals in production and respiration than in the open ocean, and atmospheric $\text{CO}_2$ uptake (Figure 4; Feely et al., 2008, 2016, 2018; Chavez et al., 2017) and, in many regions, slower than in the open ocean (Laruelle et al., 2018). Along the Pacific coast, climate-driven changes in upwelling circulation result in coastal acidification

events. As mentioned in Section 3.2, upwelling-favorable winds along this coast have intensified over recent years, especially in the northern parts of the upwelling regime (García-Reyes et al., 2015; Rykaczewski and Checkley, 2008; Rykaczewski et al., 2015; Sydeman et al., 2014). Intensified upwelling supplies deep water to the shelf that is rich in DIC and nutrients but poor in oxygen. Ocean acidification and hypoxia thus are strongly linked ecosystem stressors because low-oxygen, high-$\text{CO}_2$ conditions derive from the microbial respiration of organic matter (Feely et al., 2008, 2016, 2018). In the northern California Current

System, $p\text{CO}_2$, pH, and aragonite saturation reach levels known to be harmful to ecologically and economically important species during the summer upwelling season (Barton et al., 2012, 2015; Bednaršek et al., 2014, 2016, 2017; Feely et al., 2008, 2016; Harris et al., 2013; Sutton et al., 2016). In the Gulf of Alaska, aragonite saturation drops to near saturation values during the winter months when deep mixing occurs and surface ocean $p\text{CO}_2$ exceeds atmospheric $p\text{CO}_2$ (Evans and Mathis, 2013). Along the Pacific coast, 50% of shelf waters are projected to experience year-long undersaturation by 2050 (Gruber et al.,

2012; Hauri et al., 2013; Turi et al., 2016).

Polar regions are naturally prone to acidification because of their low temperatures (Orr et al., 2005; Steinacher et al., 2009). In many Arctic coastal regions, pH and carbonate saturation state are naturally low relative to lower-latitude coastal settings. These low levels result from higher $\text{CO}_2$ solubility, the influence of multiple sources of freshwater (e.g., riverine, glacial melt,





and sea ice melt) with varying $CO_2$ chemistries, and the high respiratory DIC content in bottom waters. The Beaufort and
Chukchi Sea continental shelves experience inflows of naturally corrosive Pacific seawater with pH as low as 7.6 (Mathis et al.,
2011). The main contributing factor to the relatively high rates of acidification in polar waters is retreating sea ice, which
adds meltwater from multiyear ice and increases the surface area of open water, thereby enhancing the uptake of atmospheric
$CO_2$ (Cai et al., 2010a; Steiner et al., 2013). These factors, in combination with increasing atmospheric $CO_2$ levels, have set a
faster pace of ocean acidification in the Arctic than projected trends in other coastal regions (Feely et al., 2009; Mathis et al.,
2015; Sutton et al., 2016). Models predict annual average aragonite undersaturation (favoring dissolution) for the Bering Sea
and the Chukchi Sea by 2070 and 2030, respectively (Mathis et al., 2015). The Beaufort Sea upper halocline and deep waters
now regularly show aragonite undersaturation (Mathis et al., 2015; Miller et al., 2014). These chemical seawater signatures
are propagated via M'Clure Strait and Amundsen Gulf into the Canadian Arctic shelf and beyond (Azetsu-Scott et al., 2010;
Turk et al., 2016; Yamamoto-Kawai et al., 2013). Model projections based on the IPCC high-$CO_2$ emissions scenario RCP 8.5,
suggest the Beaufort Sea surface water will become undersaturated with respect to aragonite around 2025 (Steinacher et al.,
2009; Steiner et al., 2014). As these conditions intensify, negative impacts for calcifying marine organisms are expected to
become a critical issue reshaping ecosystems and fisheries across the Arctic (Moore and Stabeno, 2015).

In the northern Gulf of Mexico, surface aragonite saturation states typically range from 3.6 to 4.5 and are thus well above
the thermodynamic dissolution threshold (Wang et al., 2013; Wanninkhof et al., 2015). Here excessive nutrient inputs from the
Mississippi River result in hypoxia and eutrophication-induced acidification of near-bottom waters (Cai et al., 2011; Laurent
et al., 2017). Similar to the California Current System, low-oxygen and high-$CO_2$ conditions coincide and derive from mi-
crobial respiration of organic matter (Cai et al., 2011; Laurent et al., 2017; Feely et al., 2018). Currently, aragonite saturation
states are around 2 in hypoxic bottom waters and thus well above the saturation threshold. Projections suggest that aragonite
saturation states of these near-bottom waters will drop below the saturation threshold near the end of this century (Laurent
et al., 2018).

Recent studies indicate that the northern regions of the Atlantic coast (the Mid Atlantic Bight and Gulf of Maine) are more
prone to acidification than the South Atlantic Bight (Wang et al., 2013; Wanninkhof et al., 2015). Coastal waters in this region
have, on average, lower pH and lower aragonite saturation states than more southern coastal regions. These properties are
explained primarily by a decrease in mean total alkalinity of shelf water from the SAB toward the GOM. Seasonal undersat-
uration of aragonite in subsurface water is occurring in the GOM with photosynthesis and respiration playing a major role in
controlling the seasonal variability of aragonite saturation states; dissolution of aragonite might already occur in fall and winter
(Wang et al., 2017). The GOM supports a significant shellfish industry, and displays the lowest pH and aragonite saturation
levels along the east coast in summer (Wang et al., 2013; Sutton et al., 2016).

## 5   Conclusions

Tremendous progress has been made during the last decade in improving understanding and constraining rates of carbon cycling
in coastal waters because of a greatly expanded suite of observations, process studies, and models. However, quantification of



many coastal carbon fluxes remains a significant challenge. Carbon is constantly exchanged across the air-sea interface as well as the interfaces between land and coastal ocean, coastal and open ocean waters, and water and sediment. Net exchange fluxes and trends are relatively small signals masked by a large and fluctuating background. At present, most of these fluxes are not quantified well enough to derive well-constrained carbon budgets for North American coastal waters or to project how those fluxes will change in the future due to various drivers.

This synthesis focused primarily on the role of ocean margins in sequestering atmospheric $CO_2$ and coastal ocean acidification. In the coastal ocean, a net removal of carbon from direct interaction with the atmospheric reservoir can occur by export of dissolved or particulate carbon to the deep ocean or by permanent burial in sediments. Neither of these is easily observed or well quantified. The best-observed flux is gas exchange across the air-sea interface, although extracting the small net flux and its trend from a variable background with large-amplitude seasonal fluctuations remains a challenge. Ultimately, the removal

of anthropogenic carbon is the relevant quantity for assessing the contribution of ocean margins to the uptake of anthropogenic carbon; however, the separation of anthropogenic fluxes from the natural background is thus far elusive for coastal waters. The only available estimates are from a global modeling study (Bourgeois et al., 2016).

Estimates of air-sea $CO_2$ fluxes currently provide the best evidence for the contribution of coastal waters to overall carbon uptake by the ocean. In the broad shelf system of the Atlantic coast acting as sources (e.g., near-shore regions of the SAB),

while others can be neutral or act as weak sinks (Scotian Shelf, MAB and outer SAB). Large sections of the narrow shelf of the Pacific coast are dominated by upwelling circulation, which leads to strong $CO_2$ outgassing near the coast. However, compensating for this outgassing is biologically driven uptake from upwelled nutrients further offshore. Recent estimates are consistent in suggesting that the region is a weak to moderate sink of atmospheric $CO_2$. The relatively wide shelves in the Gulf of Mexico are considered a weak net sink, with the West Florida Shelf and the western Gulf shelf acting as sources; the Mexico

shelf being neutral; and only the northern shelf a clear sink that largely is driven by anthropogenic nutrient inputs from the Mississippi River. The wide, seasonally ice-covered shelves in the North American Arctic consistently are acting as a sink for atmospheric $CO_2$. The low surface-water $pCO_2$ in this region primarily results from low water temperatures and the decreased uptake of atmospheric $CO_2$ during a significant fraction of the year because of seasonal ice cover.

Overall, North American coastal waters act as a sink of $CO_2$, but regional variations and uncertainties are large. Several

drivers influence secular trends in coastal carbon fluxes and will continue to do so in the future. These drivers include rising atmospheric $CO_2$ levels, changes in atmosphere-ocean interactions (e.g., wind forcing and heat fluxes), changes in the hydrological cycle, and anthropogenic perturbations of global nutrient cycling (particularly, the nitrogen cycle). Coastal surface $pCO_2$ does not closely track atmospheric $pCO_2$. Although there are a number of plausible mechanisms for potential future changes in coastal carbon uptake, the total effect cannot be predicted with any confidence. Regional model studies are beginning to

address these challenges.

A major concern is coastal acidification, which can affect the growth, metabolism, and life cycles of many marine organisms, specifically calcifiers, and can trigger cascading ecosystem-scale effects. Most vulnerable are those organisms that precipitate aragonite, one of the more soluble forms of biogenic $CaCO_3$ in the ocean. Aragonite saturation states are routinely below saturation (favoring dissolution) in North American Arctic coastal waters. Along the North American Pacific coast, atmospheric



$CO_2$ uptake in combination with intensified upwelling that brings low-pH, low-oxygen water onto the shelves leads to arago-
nite levels below the saturation threshold in large portions of the subsurface waters. In the northern Gulf of Mexico, aragonite
saturation states are well above the dissolution threshold. Although eutrophication-induced acidification occurs in bottom wa-
ters influenced by Mississippi River inputs of nutrients and freshwater, saturation levels remain well above the thermodynamic
dissolution threshold.

5    Given the importance of coastal margins, both in contributing to carbon budgets and in the societal benefits they provide,
further efforts to improve assessments of the carbon cycle in these regions are paramount. Critical needs are maintaining and
expanding existing coastal observing programs, continued national and international coordination, and integration of observa-
tions, modeling capabilities, and stakeholder needs.

*Data availability.* All data sets used in this synthesis are publicly available at the sources indicated.

10  *Competing interests.* The authors are not aware of any competing interests.

*Acknowledgements.* This manuscript builds on synthesis activities carried out for the 2nd State of the Carbon Cycle Report (SOCCR2). It is
also a contribution to the Marine Biodiversity Observation Network (MBON), the Integrated Marine Biosphere Research (IMBeR) project,
International Ocean Carbon Coordination Project (IOCCP), and the Cooperative Institute of the University of Miami and the National Oceanic
and Atmospheric Administration (CIMAS) under cooperative agreement NA10OAR4320143. KF was funded by the NSERC Discovery
program. SEL was funded by NASA grant NNX14AO73G. RGN was funded by NASA grant NNX14AM37G. FMK was funded through
NASA grant NNX14AP62A. This is Pacific Marine Environmental Laboratory contribution number 4837.





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




**Table 2.** Estimates of satellite-derived carbon burial and primary production (NPP) from Dunne et al. (2007), and model-simulated NPP and air-sea $CO_2$ flux from Bourgeois et al. (2016) for a decomposition of the EEZ of Canada, the U.S., and Mexico. Model estimates are calculated by averaging the years 1993–2012. Positive numbers represent fluxes into the coastal ocean. Subregion abbreviations are MAB—Mid-Atlantic Bight, GOM—Gulf of Maine, SS—Scotian Shelf, GStL—Gulf of St. Lawrence & Grand Banks, LS—Labrador Shelf, HB—Hudson Bay, CAS—Canadian Arctic shelf, BCS—Beaufort & Chukchi Seas, BS—Bering Sea, GAK-Gulf of Alaska, NCCS—Northern California Current System, CCCS—Central California Current System, SCCS—Southern California Current System, Isthmus—American isthmus, GMx—Gulf of Mexico and Yucatan Peninsula, SAB—South Atlantic Bight, Islands—Hawaii & other Pacific & Caribbean islands.

| EEZ region, area | C burial | | Satellite NPP | | Model NPP | | Model air-sea $CO_2$ flux (anthropogenic fraction) | |
|---|---|---|---|---|---|---|---|---|
| $10^3$ km$^2$ | g C m$^{-2}$ y$^{-1}$ | Tg C y$^{-1}$ | g C m$^{-2}$ y$^{-1}$ | Tg C y$^{-1}$ | g C m$^{-2}$ y$^{-1}$ | Tg C y$^{-1}$ | g C m$^{-2}$ y$^{-1}$ | Tg C y$^{-1}$ |
| MAB, 500 | 23 | 101 | 360 | 170 | 260 | 120 | 31 (14) | 15 (6.8) |
| GOM, 160 | 46 | 5.5 | 490 | 58 | 180 | 26 | 33 (7.1) | 4.9 (1.1) |
| SS, 220 | 9.8 | 2.0 | 300 | 63 | 170 | 43 | 33 (11) | 8.2 (2.8) |
| GStL, 860 | 16 | 11 | 260 | 190 | 150 | 130 | 24 (6.5) | 21 (5.6) |
| LS, 1,100 | 2.3 | 2.3 | 120 | 120 | 82 | 88 | 33 (9.5) | 36 (10) |
| HB, 1,200 | 19 | 17.1 | 144 | 130 | 130 | 150 | -0.48 (1.4) | -0.50 (1.7) |
| CAS, 1,000 | 2.6 | 1.6 | 42 | 26 | 19 | 20 | 4.1 (0.96) | 4.3 (0.96) |
| BCS, 950 | 12 | 10 | 120 | 110 | 49 | 47 | 8.0 (1.2) | 7.6 (1.1) |
| BS, 2,200 | 17 | 34 | 240 | 490 | 130 | 270 | 13 (4.0) | 28 (8.6) |
| GAK, 1,500 | 7.2 | 10.0 | 260 | 360 | 130 | 210 | 19 (4.6) | 29 (7.1) |
| CCSN, 460 | 6.1 | 2.54 | 270 | 110 | 160 | 73 | 9.4 (4.2) | 4.3 (1.9) |
| CCSC, 640 | 1.2 | 0.65 | 260 | 150 | 170 | 110 | 1.1 (4.4) | 0.72 (2.9) |
| CCSS, 1,200 | 0.99 | 1.1 | 210 | 230 | 150 | 190 | -4.3 (3.1) | -5.5 (4.0) |
| Isthmus, 1,400 | 0.42 | 0.53 | 230 | 300 | 150 | 200 | -2.3 (3.6) | -3.2 (4.9) |
| GMx, 1,600 | 6.2 | 8.7 | 250 | 350 | 220 | 360 | 4.8 (3.7) | 7.9 (6.2) |
| SAB, 500 | 5.4 | 2.4 | 210 | 92 | 260 | 130 | 9.7 (6.6) | 5.0 (3.4) |
| Islands, 7,500 | .0055 | .041 | 120 | 890 | 80 | 620 | -1.4 (4.1) | -11 (31) |
| Total | - | 120 | - | 3,400 | - | 2,800 | - | 150 (100) |
| Total w/o Islands | - | 120 | - | 2,500 | - | 2,200 | - | 160 (59) |