# Peer review of "Carbon cycling in the North American coastal ocean: A synthesis"

_Biogeosciences, 2018_

## Short Comment (SC1) · 1 Nov 2018

*A note upfront from the submitting person: I am a master students in earth system science at the University of Zurich. The review was part of an exercise during a master level seminar. I would like to highlight that the depth of scientific knowledge and technical understanding of this review represents that of master student. I enjoyed discussing the manuscript in the seminar, and hope that our comments will be helpful for the authors.*

The authors did a review to summarize the recent findings of costal carbon uptake and ocean acidification for the margins of North America. It was a part of an assessment for the 2nd State of the Carbon Cycle report (SOCCR-2). The following research questions

were asked: "1) whether the costal ocean of North America takes up atmospheric CO2 and subsequently exports it to the deep ocean, and 2) discuss patterns and drivers of coastal ocean acidification". In a first step the authors give an overview on the different carbon stocks that exist in the coastal waters and the mechanics that moves the carbon pools from one to another. Then the synthesis looks at the stock in different areas around the North American margin and their carbon fluxes. Fennel et al. also look at the overall up take of CO2 in the North American margins and the influence of the anthropogenic CO2 on it. Furthermore, the authors showed the acidification trend and the main driver for it. Fennel et al. concluded that in general the North American costa acts as a sink for atmospheric carbon. However, the authors also mention that there are large uncertainties and reginal variation. As for the acidification Fennel et al. concluded, that the North American costal water are below aragonite saturation and therefore, favoring dissolution. In general the synthesis gives an overview of the work that is done on the carbon fluxes and stocks around the North American costal ocean and therefore answers the question about the Carbon cycling in this area. The data that is shown in the review support the conclusion that Fennel at al. have made. The synthesis gives a good overview of the work that is done, however I have some concerns with this study:

The main concern is that there is no description of the approach to the review. For me it was not clear how the data was generated for Sections 3 and 4. I would suggest to see and follow the suggestions in the paper by Gurevitch et al. (2018) on how to do a systematic Review. These authors identified the four stages of the systematic review process ('identification', 'screening', 'eligibility' and 'included'). I think if this this would be included in the review, it would help to improve it significantly because then the reader would know how the papers that were used in this synthesis were selected and analyzed, and it would make the review more reproducible. It would also highlight how complete is your review and give strength to the findings and gaps in knowledge identified.

In my opinion Sections 2, 3 and 4 need some additional editing. Section 2 is confusing and needs some focus. It seems to me that it provides too many topics and the flow between them could be improved. In general, this section could go from the general overview to a more detailed view of the carbon fluxes, even consider removing the detailed mechanistic explanations and focus on sections 3 and 4 which are the ones that address your research question. This adjustment would improve the paper because it makes it easier for the reader to follow the study. Section 3 (page 6-17) is written quite differently from the rest of the text and this makes it difficult to follow the argument. For example Section 3.2 and 3.3 have the same structure and both section are clear verbalized so that the reader follows the main arguments easily. This is not the case for section 3.1 and 3.4. Section 4 (page 17): In all the other sections there was a short introduction before the subsection were addressed. I would recommend to do this here as well, because this would lead to a consistency in your paper.

In my opinion Section 3.4 (page 13-14) fails to address its stated purpose, and perhaps one of the goals of the review. This section was stated to review the fluxes that take place in north American oceans, but instead it does not mention the numbers of the fluxes that are shown in figure 3 (page 7) and gives more of an overview of how the fluxes mechanics are in the arctic region. I would suggest to remove general introduction to mechanisms and add more of an overview of the different fluxes.

Minor comments:

Page 1 – Is Figure 1 out of place? Why not combine it with Figure 3?

Page 3 lines 10-20 – I find this information too detailed for the introduction and would suggest to move it to section three or to a new section methods and study area section.

Page 4 line 3-15 – There are no references in these sections. I did not understand if this is a conclusion of yours or not. If not, could you provide some references?

Page 5 line 14-22 – There are no references in these sections I did not understand if

this is a conclusion of yours or not. If not, could you provide some references?

Page 6 line 2-5 – There are no references in these sections I did not understand if this is a conclusion of yours or not. If not, could you provide some references?

Page 6 line 8-14 – This paragraph looks out of context here since it does not connect with the rest of this section. Maybe this part (page 6 line 8-14) could be added to page 2 line 3-7.

Page 6 line 18-19 – the authors state that the fluxes might not be comparable. I would suggest to add some information on how you interpreted these variable estimates from different methods.

Page 6 line 21-25 – There are no references in these sections I did not understand if this is a conclusion of yours or not. If not, could you provide some references?

Page 11 line 30 – is the unit of the 14 Tg C sink correct? If a flux shouldn't it be per unit of time? Figure 6 (page 19) – Why not making the figure a bit bigger?

Why is table 2 (page 36) is after the references? My guess is that the files got mixed up during the uploading.

References: Gurevitch, J., Koricheva, J., Nakagawa, S. and Stewart, G. (2018). Meta-analysis and the science of research synthesis. Nature, 555(7695), 175.

---

## Referee Comment (RC1) · GRUBER (Referee) · 4 Jan 2019

**1  Summary**

Fennel et al. assess and synthesize a large suite of observation- and model-based studies in order to derive regionally specific carbon budgets for the coastal regions around North America, with a special focus on the air-sea $CO_2$ flux. They find that the entire North American coastal ocean acts as a sink for atmospheric $CO_2$, currently taking up about $160 \pm 80$ Tg C $yr^{-1}$. The majority of this total sink stems from the high latitude regions, as the sink strength of the temperate and subtropical latitudes is relatively weak. Fennel et al. also attempt to assess trends in these fluxes and in ocean acidification, but the number of studies available was too small to draw clear

conclusions.

**2 Evaluation**

Even though the potentially substantial role of the coastal ocean to the global ocean carbon sink is well recognized, the quantification of the carbon fluxes in coastal regions has remained a large challenge. Despite a large increase in the number of observations and modeling studies, the data remain sparse in comparison to the large spatial and temporal variability that characterize these regions. Given these challenges, it is no surprise that the majority of the published studies focus on a single region or a small set of regions, with only a handful of studies having attempted to construct a larger scale synthetic view of the coastal ocean.

This is the opportunity that this study capitalizes on. It assembled a large suite of local to regional studies in the coastal regions around North America, assessed them, and synthesized them into a set of coherent estimates for the entire continent. Thus, such a synthesis is a highly welcomed addition to the scientific body of literature. Even though - by definition - there is only a limited amount of truly novel information in such a review, the synthetic aspect of this work is very important and fully justifies the potential publication of this manuscript. I commend the authors for their detailed efforts to bring the different studies together and to put them into a common framework for discussion and synthesis. The approach taken, and the conclusions drawn from them are clear and follow standard protocols. The manuscript is overall well written, with a set of clear messages. I am thus very supportive of the publication of this manuscript. But in addition to several specific points listed below, I have two major comments that I would like the authors to consider before giving my final ok:

- (i) *Length, Scope, and overall balance*: The paper is quite long and at the same time unbalanced across the different parts in terms of depths and insight. Overall,

it feels as if the paper was taken in relatively unmodified form from the SOCCR2 report without thinking too much about the opportunity that the publication in a peer-reviewed journal would provide in order to push the synthesis and discussion further. I thus highly recommend to revisit the paper with fresh eyes, consider its scope, and to rewrite/edit it vigorously with the aim to enhance its impact.

For example: The first part of the paper contains a lot of rather basic review elements. Some of this material is certainly needed, but many of the things that are described are rather basic and often not even used later in the assessments. I thus suggest to prune this section rather strongly. This should be taken also as an opportunity to enhance the referencing in this section.

In contrast: The discussion is relatively brief, missing the opportunity for bringing new knowledge to the table. For example, the air-sea $CO_2$ flux estimates are not put into the context of the fluxes in the adjacent open ocean. Are these flux densities unusual compared to the open ocean? As far as I can tell, both the magnitude and pattern of the flux densities in the coastal ocean are, in fact, not that different from those we see globally across the whole ocean. I think this is worth discussing. Also, there is much interest to better understand the role of the coastal ocean as part of the "aquatic continuum", i.e., the lateral transport of carbon (and nutrients) from the land via river systems to the estuaries, and then through the coastal systems into the open ocean. This is briefly touched upon in the paper, but not further elaborated. Finally, it would be interesting for the average reader to get a sense what the $160 \pm 80$ Tg C yr$^{-1}$ means compared to other important carbon fluxes, e.g., fossil emissions over North America, uptake by the North American continent, etc.

I also suggest to strengthen the conclusions. I think it would be good to know from this distinguished set of researchers what they recommend in terms of research/observational efforts. Which research aspect should be strengthened, which new types of observation should be put in place, or which impact should

be studied in more detail?

- (ii) *Synthesis*: This concerns the approach taken in section 3.5 in order to arrive at a synthetic view of the North American coastal air-sea $CO_2$ fluxes. The authors decided - after having done a fantastic job to synthesize the regional fluxes - to mostly forgo the regional studies and rely primarily on the model study of Bourgeois et al. to arrive at a best estimate for the whole continent. This is a wasted opportunity, in my opinion. I think the authors should bring forward the local studies much more prominently and attempt to truly synthesize the more regional studies with those that take a more top-down approach. Concretely, I suggest to replace the Chen et al. (2013) numbers in table 1 (they upscaled a few point studies only) with a summary from the regional studies shown in Figure 3 (Table S1) and then arrive at a synthesis estimate by combining the three sources of information, i.e., the SOCATv2 based estimate by Laruelle, the model based estimate by Bourgeois, and the regional estimates from this study. This would give this work much more impact in the end.

**3  Recommendation**

I recommend to accept this manuscript after moderate revision. The core of the manuscript is solid and well described, but there is substantial potential for enhancing the impact of this paper by strengthening the discussion and conclusion sections at the expense of the background section.

**4   Specific comments**

Abstract line 10: "not well constrained": It is unclear whether this is a statement about the formal uncertainty estimate ($\pm$ 80 Tg C yr$^{-1}$ or whether this is a statement of confidence, i.e., how uncertain are the authors about this uncertainty statement.

Abstract, general: The abstract is written in a relatively general and rather descriptive manner. It would benefit greatly from being more quantitative, and from making the link to the underlying processes.

page 3, lines 10 to 20: This set of clear definition is very helpful. Yet later on, when the different studies are compared and assessed, it is not always clear how the originally reported numbers were then rescaled/transformed to refer just to the EEZ.

page 4, Figure 2 and text: The perspective taken here is one that is almost purely one-dimensional in lateral direction, whereas in reality, alongshore transports and the vertical structure of the exchange with the adjacent open ocean matter a great deal (see e.g., Frischknecht et al. 2018). This is not that relevant for the air-sea $CO_2$ flux, but matters greatly for any discussion about the nature and magnitude of the lateral exchange of inorganic and organic carbon within the ocean.

pages 6-17, entire section 3: As mentioned above, it would be helpful to know how the numbers from the original studies were re-scaled or transformed into the numbers reported and discussed in this section.

pages 6-17, entire section 3: It would help the reader if the individual sections were structured in a similar manner. While the air-sea $CO_2$ flux is clearly always in the center, the different sections treat the other elements, such as the processes explaining the fluxes, and in particular, the lateral fluxes of organic and inorganic carbon very differently.

page 6, line 10: "The anthropogenic component of a given carbon flux is defined as

the difference between its preindustrial and present-day fluxes." This definition of the anthropogenic carbon component differs in an important manner from that adopted for global ocean carbon studies (see e. g., McNeil and Matear (2013).) With this very broad definition, the authors include also changes induced by natural processes as "anthropogenic". In contrast, the open ocean community tends to define the anthropogenic carbon component as just this part of the flux/change in storage that occurs as a result of the increase in atmospheric $CO_2$. I am painfully aware that there is no generally accepted definition, but I would argue that the chosen, very broad definition of "anthropogenic" is in the end more confusing than illuminating. This is particularly the case since the Bourgeois et al. paper used to constrain the anthropogenic $CO_2$ component follows the more classical open ocean definition (see there comment on page 4171).

page 7, Figure 3: I very much like this figure. I suggest to color code the different estimates to emphasize the very different types of estimates. For example, by differentiating between model- and observation-based estimates. And then one could further differentiate between those that really represent a long-term mean flux, versus those that pertain to a single year only, etc.

page 12, line 9: You may want to consider adding the Frischknecht et al. (2018) study here. They show that about a third of the organic matter produced in the first 100 km gets transported offshore.

page 13, lines 24-27: This paragraph does not fit here. I recommend to move it to the trends section.

page 14, section 3.5: I urge the authors to reconsider their decision to downplay their regional estimates that much (see my second major comment above).

page 15, figure 5 and text: I very much like this synthesis of the fluxes. But I also think that the authors need to caveat this in an important manner. First, the uncertainties are probably underestimated given the unrepresented error contribution from temporal

variability and other systematic error sources. Second, this implicitly assumes some form of steady-state. And third, there are probably very large regional differences hidden beneath this continental summary. It is probably worth to open the Pandora box a bit with regard to the regional aspects here, since I would expect that it is primarily the Arctic/high latitude regions that contribute to this net offshore transport.

page 17-18: Section 4: Relative to the previous section 3, this section could benefit from some better structuring. It is also not so clear what message the authors want to convey except that "things are inconclusive". I think that more is possible here.

page 17, lines 20 through 23: "by definition, changes in total carbon fluxes imply changes in anthropogenic fluxes as well." See my comment above about the definition of anthropogenic $CO_2$. I suggest to reconsider this very broad definition of "anthropogenic $CO_2$".

page 18, line 6: "pCO2 has increased faster than in the open ocean due to the combined effects of atmospheric uptake..." What is the evidence for this process being important? Such a behavior is not easy to achieve in a system that is generally outgassing - here, only a reduction in the surface residence time could achieve this. Thus, the much more important process explaining the higher than expected trend is ocean circulation and biology.

page 18, lines 10ff: "Some studies suggest that trends in the air-sea $pCO_2$ gradient ($\Delta pCO_2$) are indicative of a strengthening or weakening of the net CO2 uptake by shelf systems...". This is not quite correctly formulated. A trend in $\Delta pCO_2$ does not "indicate" a trend in the air-sea $CO_2$ flux, but actually "implies" a trend.

page 18, lines 20 through 26: I don't agree with the line of arguments here (or at least how I understood them). The authors seem to suggest that because the anthropogenic trend cannot be detected before 30 years have passed, the trends by Laruelle are likely not real. In my opinion, the discussion needs to be structured much more carefully. First, the trends detected by Laruelle et al. (2018) cover a rather short period, and thus are much more likely the result of internal variability than the result of an anthropogenic (forced) trend (in climate). Second, the time of emergence described by McKinley et al. (2016) checks for the emergence of a forced climate change signal versus that driven by internal variability. This does not imply that there is no trend to be detected, for example, in the Delta pCO2, if this trend was solely caused by the increase in atmospheric CO2, i.e., the uptake of anthropogenic $CO_2$ (sensu strictu). Please reconsider the writing of this paragraph.

page 19, lines 4 and 5: I don't know of any occasion where the impact of the changes in wind-stress on the gas transfer velocity had been more important than the impact on the ocean circulation. Thus, this sequence, i.e. gas exchange first, circulation second, does not reflect the common situation.

page 19ff, section 4.2: As section 4.1, this section could benefit from some better structuring. It largely consists of some general statements about ocean acidification, but does not really advance the knowledge much. It would greatly benefit from some more quantitative assessments of the distribution and the trends in OA.

page 20, line 4: "...biological production and respiration". Even more important is ocean transport and mixing. (see the next sentence)

page 20, line 19: "events". Here is one example of why a better structuring of this section is needed. This sentence starts without context, and as a reader one is baffled about the expression "events", when the subject is long-term trend. etc.

page 20, line 21, "Polar regions are naturally prone to acidification because of their low temperatures" This sentence is problematic for two reasons. First, the polar regions are not really prone to acidification per se (there sensitivity to change is actually lower, e.g., del pH/del pCO2), but what makes them prone is them being much closer to critical thresholds. Second, this sentence implies that their low pH or low Omega is a result of the waters being cold. This is not really correct either. It is because these waters have a high DIC/Alk ratio. And yes, the low temperatures are one of the reasons for the

high DIC/Alk ratios, but also the freshwater inputs, the much more common mixing with deeper high DIC/Alk waters (owing to remineralization), and their generally low Alk are important factors. This is readily illustrated if one contrasts the pH and Omega values in the Atlantic to the Pacific at the same temperature.

page 21, lines 34ff: "The main contributing factor to the relatively high rates of acidification in polar waters is retreating sea ice, which adds meltwater from multiyear ice and increases the surface area of open water, thereby enhancing the uptake of atmospheric CO2 (Cai et al., 2010a; Steiner et al., 2013)." This sentence is another good example for the need to better structure this section. It makes a statement about the cause of the faster trend for OA in the Arctic even before a statement has been made that the trends are actually larger. Thus, across the board in this section, it behooves the authors well to first establish a good base line of the expected trends, then report the observed trends and only then start to discuss the potential drivers for these trends.

page 21, lines 21ff: "more prone". See my comment above. It would be good to be more specific what is meant here.

page 21, Conclusions. Before advancing to the conclusion section, it would strengthen the manuscript substantially if the authors expanded the discussion section by considering additional material (see my first major comment above). Examples, are coastal versus open ocean, comparison of fluxes across compartments, and recommendation for future studies, etc.

page 22: line 9-10: "Ultimately, the removal of anthropogenic carbon is the relevant quantity for assessing the contribution of ocean margins to the uptake of anthropogenic carbon". This is confusing - what is the difference between "removal" and "uptake"? Please explain better what is meant here. Is it global versus coastal?

**5 Cited References**

McNeil, B. I., and R. J. Matear (2013), The non-steady state oceanic $CO_2$ signal: its importance, magnitude and a novel way to detect it, Biogeosciences, 10(4), 2219–2228, doi:10.5194/bg-10-2219-2013.

Frischknecht, M., M. Münnich, and N. Gruber (2018), Origin, Transformation, and Fate: The Three-Dimensional Biological Pump in the California Current System, J. Geophys. Res. Ocean., 123(11), 7939–7962, doi:10.1029/2018JC013934.

Nicolas Gruber, January 2019

––––––––––––––––––––––––––––––

---

## Referee Comment (RC2) · Rabouille (Referee) · 7 Jan 2019

The paper presents a synthesis of carbon cycling on the continental margins of the United States , including the Atlantic, the Pacific and the Arctic coasts. The authors concentrate on the CO2 exchange between air and seawater as it is the most documented and tabulated parameter for carbon exchange in the ocean, and specifically in coastal zones. They propose an overall map of air-sea CO2 exchange for the East, West and Arctic coasts which shows a large tendency of these shelves to be a sink for CO2 (Fig. 3) and then discuss the results by geographic zone which makes a long and rather tedious paper to read. At the end, the acidification question is raised and shortly discussed. Overall, the paper is long but well written and proposes a good synthesis for all researcher interested by carbon cycling in the coastal zone. By discussing the

different regions separately and addressing different "cases" (large shelves, upwelling, arctic, enclosed sea-GoM), they provide insightful explanation for the observed CO2 fluxes. I think that the paper is publishable after minor revisions.

Detailed comments:

Page 3, line 10: The coastal ocean is defined as "non-estuarine waters", which is questionable regarding CO2 budgets for the coastal zone since estuaries and deltas are sources of similar magnitude compared to coastal ocean sink. . . It should be emphasized at this point in text that this cutting off estuaries and deltas is because of the paper by Najjar et al. (2018) which has already addressed the estuarine and tidal wetland's part of the C Cycle.

Page 5, line 15: "colder shelf water is denser". The authors should cite reference work by Canals et al. (2006, Nature Vol 444, doi:10.1038/nature05271) on cascading in canyons which is the ultimate case of cold and dense water diving in the lighter open sea.

Page 6, line 8: "burial and export of carbon. . . remove atmospheric CO2". The authors should specify that this is true only for ORGANIC carbon, as inorganic carbon burial will result in net CO2 increase in the zone of CaCO3 formation (2 HCO3- + Ca2+ → CaCO3 + CO2 + H2O).

Page 14, line17-18: "DIC is a component of the brine whereas total alkalinity precipitates. . .as ikaite". I do not understand why alkalinity would precipitate without DIC. He sentence should be modified to "precipitation of CaCO3 as ikaite changes the DIC to Alkalinity ratio by consuming two times more Alk than DIC during the process. . ."

Page 18, line 10: ". . . pCO2 gradient are indicative" I would rather say "is the appropriate indicator for"

Page 18, line 12: "and cross shelf exchange", please add "potential" cross shelf exchange

Page 19, Fig. 6 legend: please add uncertainty to the slopes "1.86 ppm/y and 1.95 ppm/y " as reported in text for the shelf seas (pge 18, line 14-17).

Page 20, line 10: cut sentence after "oyster larvaes in the coastal Pacific Ocean".

Page 22, line 13-16: These sentences which describe the overall CO2 flux in the coastal ocean make little sense. The main conclusion of the study is that these margins are sinks for CO2 (either natural or anthropogenic) and this should be the first and main sentence in this paragraph, before summarizing the details.

―――――――――――――――――

---

## Author Comment (AC1) · 28 Jan 2019

**Response to the interactive comment by Sandra Werthmueller**

Below the entire review is pasted in black and responses are in blue font.

*A note upfront from the submitting person: I am a master students in earth system science at the University of Zurich. The review was part of an exercise during a master level seminar. I would like to highlight that the depth of scientific knowledge and technical understanding of this review represents that of master student. I enjoyed discussing the manuscript in the seminar, and hope that our comments will be helpful for the authors.*

The authors did a review to summarize the recent findings of costal carbon uptake and ocean acidification for the margins of North America. It was a part of an assessment for the 2nd State of the Carbon Cycle report (SOCCR-2). The following research questions were asked: "1) whether the costal ocean of North America takes up atmospheric CO2 and subsequently exports it to the deep ocean, and 2) discuss patterns and drivers of coastal ocean acidification". In a first step the authors give an overview on the different carbon stocks that exist in the coastal waters and the mechanics that moves the carbon pools from one to another. Then the synthesis looks at the stock in different areas around the North American margin and their carbon fluxes. Fennel et al. also look at the overall up take of CO2 in the North American margins and the influence of the anthropogenic CO2 on it. Furthermore, the authors showed the acidification trend and the main driver for it. Fennel et al. concluded that in general the North American costa acts as a sink for atmospheric carbon. However, the authors also mention that there are large uncertainties and reginal variation. As for the acidification Fennel et al. concluded, that the North American costal water are below aragonite saturation and therefore, favoring dissolution. In general the synthesis gives an overview of the work that is done on the carbon fluxes and stocks around the North American costal ocean and therefore answers the question about the Carbon cycling in this area. The data that is shown in the review support the conclusion that Fennel at al. have made. The synthesis gives a good overview of the work that is done, however I have some concerns with this study:

The main concern is that there is no description of the approach to the review. For me it was not clear how the data was generated for Sections 3 and 4. I would suggest to see and follow the suggestions in the paper by Gurevitch et al. (2018) on how to do a systematic Review. These authors identified the four stages of the systematic review process ('identification', 'screening', 'eligibility' and 'included'). I think if this this would be included in the review, it would help to improve it significantly because then the reader would know how the papers that were used in this synthesis were selected and analyzed, and it would make the review more reproducible. It would also highlight how complete is your review and give strength to the findings and gaps in knowledge identified.

**Response:** All published regional studies that report carbon fluxes along the Atlantic, Pacific, Arctic and Gulf of Mexico coasts up to mid-2018 are included. Thus, we don't share the concern about reproducibility. If the Ms. Wertheimer is aware of a relevant study that was omitted, we would be interested to know.

In my opinion Sections 2, 3 and 4 need some additional editing. Section 2 is confusing and needs some focus. It seems to me that it provides too many topics and the flow between them could be improved. In general, this section could go from the general overview to a more detailed view of the carbon fluxes, even consider removing the detailed mechanistic explanations and focus on sections 3 and 4 which are the ones that address your research question. This adjustment would improve the paper because it makes it easier for the reader to follow the study. Section 3 (page 6-17) is written quite differently from the rest of the text and this makes it difficult to follow the argument. For example Section 3.2 and 3.3 have the same structure and both section are clear verbalized so that the reader follows the main arguments easily. This is not the case for section 3.1 and 3.4. Section 4 (page 17): In all the other sections there was a short introduction before the subsection were addressed. I would recommend to do this here as well, because this would lead to a consistency in your paper.

**Response:** The manuscript will be revised.

In my opinion Section 3.4 (page 13-14) fails to address its stated purpose, and perhaps one of the goals of the review. This section was stated to review the fluxes that take place in north American oceans, but instead it does not mention the numbers of the fluxes that are shown in figure 3 (page 7) and gives more of an overview of how the fluxes mechanics are in the arctic region. I would suggest to remove general introduction to mechanisms and add more of an overview of the different fluxes.

**Response:** This section will be revised.

Minor comments:

Page 1 – Is Figure 1 out of place? Why not combine it with Figure 3?

**Response:** No action.

Page 3 lines 10-20 – I find this information too detailed for the introduction and would suggest to move it to section three or to a new section methods and study area section.

**Response:** This contrasts with Reviewer 1's comment about this paragraph. Will leave here.

Page 4 line 3-15 – There are no references in these sections. I did not understand if this is a conclusion of yours or not. If not, could you provide some references?

**Response:** No action.

Page 5 line 14-22 – There are no references in these sections I did not understand if this is a conclusion of yours or not. If not, could you provide some references?

**Response:** No action.

Page 6 line 2-5 – There are no references in these sections I did not understand if this is a conclusion of yours or not. If not, could you provide some references?

**Response:** No action.

Page 6 line 8-14 – This paragraph looks out of context here since it does not connect with the rest of this section. Maybe this part (page 6 line 8-14) could be added to page 2 line 3-7.

**Response:** Will consider this.

Page 6 line 18-19 – the authors state that the fluxes might not be comparable. I would suggest to add some information on how you interpreted these variable estimates from different methods.

**Response:** Will be done (see responses to Reviewer 1).

Page 6 line 21-25 – There are no references in these sections I did not understand if this is a conclusion of yours or not. If not, could you provide some references?

**Response:** No action.

Page 11 line 30 – is the unit of the 14 Tg C sink correct? If a flux shouldn't it be per unit of time? Figure 6 (page 19) – Why not making the figure a bit bigger?

**Response:** Yes, the unit should be per year. Figure will be made a bit bigger.

Why is table 2 (page 36) is after the references? My guess is that the files got mixed up during the uploading.

**Response:** A Latex quirk. Table will be placed properly during typesetting.

References: Gurevitch, J., Koricheva, J., Nakagawa, S. and Stewart, G. (2018). Meta- analysis and the science of research synthesis. Nature, 555(7695), 175.

---

## Author Comment (AC3) · 28 Jan 2019

**Response to the interactive comment by Reviewer 2 (Christophe Rabouille)**

Below the entire review is pasted in black and responses are in blue font.

The paper presents a synthesis of carbon cycling on the continental margins of the United States , including the Atlantic, the Pacific and the Arctic coasts. The authors concentrate on the CO2 exchange between air and seawater as it is the most documented and tabulated parameter for carbon exchange in the ocean, and specifically in coastal zones. They propose an overall map of air-sea CO2 exchange for the East, West and Arctic coasts which shows a large tendency of these shelves to be a sink for CO2 (Fig. 3) and then discuss the results by geographic zone which makes a long and rather tedious paper to read. At the end, the acidification question is raised and shortly discussed. Overall, the paper is long but well written and proposes a good synthesis for all researcher interested by carbon cycling in the coastal zone. By discussing the different regions separately and addressing different "cases" (large shelves, upwelling, arctic, enclosed sea-GoM), they provide insightful explanation for the observed CO2 fluxes. I think that the paper is publishable after minor revisions.

**Response:** We would like to thank Christophe Rabouille for reviewing our manuscript and for providing his constructive and positive comments. Addressing these and Reviewer 1's comments will make for a much improved presentation. Envisioned changes are detailed below.

Detailed comments:

Page 3, line 10: The coastal ocean is defined as "non-estuarine waters", which is questionable regarding CO2 budgets for the coastal zone since estuaries and deltas are sources of similar magnitude compared to coastal ocean sink. . . It should be emphasized at this point in text that this cutting off estuaries and deltas is because of the paper by Najjar et al. (2018) which has already addressed the estuarine and tidal wetland's part of the C Cycle.

**Response:** Agree. In the revision more information on estuarine waters will be added while, at the same time, emphasizing the reason for the cut off.

Page 5, line 15: "colder shelf water is denser". The authors should cite reference work by Canals et al. (2006, Nature Vol 444, doi:10.1038/nature05271) on cascading in canyons which is the ultimate case of cold and dense water diving in the lighter open sea.

**Response:** Yes, thank you.

Page 6, line 8: "burial and export of carbon. . . remove atmospheric CO2". The authors should specify that this is true only for ORGANIC carbon, as inorganic carbon burial will result in net CO2 increase in the zone of CaCO3 formation (2 HCO3- + Ca2+ → CaCO3 + CO2 + H2O).

**Response:** Agree. Will be done.

Page 14, line17-18: "DIC is a component of the brine whereas total alkalinity precipitates. . .as ikaite". I do not understand why alkalinity would precipitate without DIC. He sentence should be modified to "precipitation of CaCO3 as ikaite changes the DIC to Alkalinity ratio by consuming two times more Alk than DIC during the process. . ."

**Response:** Agree. Will be done.

Page 18, line 10: ". . . pCO2 gradient are indicative" I would rather say "is the appropriate indicator for"

**Response:** Prefer to change simply to "indicate."

Page 18, line 12: "and cross shelf exchange", please add "potential" cross shelf exchange

**Response:** Adding "potential" would change the meaning in a way that is not intended.

Page 19, Fig. 6 legend: please add uncertainty to the slopes "1.86 ppm/y and 1.95 ppm/y " as reported in text for the shelf seas (pge 18, line 14-17).

**Response:** Will be done.

Page 20, line 10: cut sentence after "oyster larvaes in the coastal Pacific Ocean".

**Response:** Will break into two or more sentences.

Page 22, line 13-16: These sentences which describe the overall CO2 flux in the coastal ocean make little sense. The main conclusion of the study is that these margins are sinks for CO2 (either natural or anthropogenic) and this should be the first and main sentence in this paragraph, before summarizing the details.

**Response:** Will be revised.

---

## Author Response (AR1)

**Response to the interactive comment by Reviewer 1 (Niki Gruber)**

Below the entire review is pasted in black and responses are in blue font.

**1 Summary**

Fennel et al. assess and synthesize a large suite of observation- and model-based studies in order to derive regionally specific carbon budgets for the coastal regions around North America, with a special focus on the air-sea $CO_2$ flux. They find that the entire North American coastal ocean acts as a sink for atmospheric $CO_2$, currently taking up about $160 \pm 80$ Tg C $yr^{-1}$. The majority of this total sink stems from the high latitude regions, as the sink strength of the temperate and subtropical latitudes is relatively weak. Fennel et al. also attempt to assess trends in these fluxes and in ocean acidification, but the number of studies available was too small to draw clear conclusions.

**Response:** We would like to thank Niki Gruber for carefully reviewing our manuscript and providing his comprehensive, constructive and encouraging comments. Addressing these in the revision processes resulted in a much-improved presentation. Point-by-point responses and resulting changes are given below.

**2 Evaluation**

Even though the potentially substantial role of the coastal ocean to the global ocean carbon sink is well recognized, the quantification of the carbon fluxes in coastal regions has remained a large challenge. Despite a large increase in the number of observations and modeling studies, the data remain sparse in comparison to the large spatial and temporal variability that characterize these regions. Given these challenges, it is no surprise that the majority of the published studies focus on a single region or a small set of regions, with only a handful of studies having attempted to construct a larger scale synthetic view of the coastal ocean.

This is the opportunity that this study capitalizes on. It assembled a large suite of local to regional studies in the coastal regions around North America, assessed them, and synthesized them into a set of coherent estimates for the entire continent. Thus, such a synthesis is a highly welcomed addition to the scientific body of literature. Even though - by definition - there is only a limited amount of truly novel information in such a review, the synthetic aspect of this work is very important and fully justifies the potential publication of this manuscript. I commend the authors for their detailed efforts to bring the different studies together and to put them into a common framework for discussion and synthesis. The approach taken, and the conclusions drawn from them are clear and follow standard protocols. The manuscript is overall well written, with a set of clear messages. I am thus very supportive of the publication of this manuscript. But in addition to several specific points listed below, I have two major comments that I would like the authors to consider before giving my final ok:

• (i) *Length, Scope, and overall balance*: The paper is quite long and at the same time unbalanced across the different parts in terms of depths and insight. Overall, it feels as if the paper was taken in relatively unmodified form from the SOCCR2 report without thinking too much about the opportunity that the publication in a peer-reviewed journal would provide in order to push the synthesis and discussion further. I thus highly recommend to revisit the paper with fresh eyes, consider its scope, and to rewrite/edit it vigorously with the aim to enhance its impact.

For example: The first part of the paper contains a lot of rather basic review elements. Some of this material is certainly needed, but many of the things that are described are rather basic and often not even used later in the assessments. I thus suggest to prune this section rather strongly. This should be taken also as an opportunity to enhance the referencing in this section.

In contrast: The discussion is relatively brief, missing the opportunity for bringing new knowledge to the table. For example, the air-sea $CO_2$ flux estimates are not put into the context of the fluxes in the adjacent open ocean. Are these flux densities unusual compared to the open ocean? As far as I can tell, both the magnitude and pattern of the flux densities in the coastal ocean are, in fact, not that different from those we see globally across the whole ocean. I think this is worth discussing. Also, there is much interest to better understand the role of the coastal ocean as part of the "aquatic continuum", i.e., the lateral transport of carbon (and nutrients) from the land via river systems to the estuaries, and then through the coastal systems into the open ocean. This is briefly touched upon in the paper, but not further elaborated. Finally, it would be interesting for the average reader to get a sense what the $160 \pm 80$ Tg C $yr^{-1}$ means compared to other important carbon fluxes, e.g., fossil emissions over North America, uptake by the North American continent, etc.

**Response:** The positive assessment is much appreciated. In response to the four specific suggestions:

1) "revisit the paper with fresh eyes, consider its scope, and rewrite/edit it vigorously with the aim to enhance its impact"—The manuscript was reviewed with fresh eyes and several sections were vigorously edited (e.g. section 2, 3.4, 3.5, 4.1, 4.2, 5). Section 2 was shortened and Figure 2 was removed.

2) "air-sea $CO_2$ flux estimates are not put into the context of the fluxes in the adjacent open ocean" – Section 3.5 was edited and a new figure was added. Also see more detailed response below.

3) "the "aquatic continuum" [..] is briefly touched upon in the paper, but not further elaborated" – Actually, the term "aquatic continuum" does not occur in the manuscript at all. This is a deliberate choice because this review is focused on continental shelves only and explicitly avoids discussion of freshwater, estuarine and wetland fluxes. These are discussed in separate reviews but would be highly relevant to the coastal continuum. The study by Regnier et al. (2013) is cited a few times where appropriate, but without reference to the continuum. The limitation in scope to continental shelf fluxes was clarified in section 1.

4) "it would be interesting for the average reader to get a sense what the $160 \pm 80$ Tg C $yr^{-1}$ means compared to other important carbon fluxes" – Agree. This flux is now compared to the global estimated ocean uptake of 2500 Tg C $yr^{-1}$ in the abstract and conclusions.

I also suggest to strengthen the conclusions. I think it would be good to know from this distinguished set of researchers what they recommend in terms of research/observational efforts. Which research aspect should be strengthened, which new types of observation should be put in place, or which impact should be studied in more detail?

**Response:** The conclusion section was strengthened and made more quantitative. However, it was decided to not recommend specific research efforts and instead endorse the broad, general approach of sustaining and expanding existing coastal observing programs, continue national and international coordination, and increase integration of observations, modeling capabilities, and stakeholder needs.

• (ii) *Synthesis*: This concerns the approach taken in section 3.5 in order to arrive at a synthetic view of the North American coastal air-sea $CO_2$ fluxes. The authors decided - after having done a fantastic job to synthesize the regional fluxes - to mostly forgo the regional studies and rely primarily on the model study of Bourgeois et al. to arrive at a best estimate for the whole continent. This is a wasted opportunity, in my opinion. I think the authors should bring forward the local studies much more prominently and attempt to truly synthesize the more regional studies with those that take a more top-down approach. Concretely, I suggest to replace the Chen et al. (2013) numbers in table 1 (they upscaled a few point

studies only) with a summary from the regional studies shown in Figure 3 (Table S1) and then arrive at a synthesis estimate by combining the three sources of information, i.e., the SOCATv2 based estimate by Laruelle, the model based estimate by Bourgeois, and the regional estimates from this study. This would give this work much more impact in the end.

**Response:** A number of edits were made to this section to draw more on the regional studies and clarify the motivation for the synthesis approach. The Table was modified as follows. The Chen et al. (2013) estimates have been removed from Table 1 for the reason suggested. With regard to adding the regional estimates from Fig. 3/Table S1 to Table 1, this didn't seem like a good option because this Table uses the coarse MARCATS segments rather than the finer grained EEZ decomposition used in Table 2. Adding the regional estimates to Table 2 was considered but deemed impractical because it would have resulted in an unwieldy table only to duplicate information from Table S1 and Figure 3. Instead, a new Figure was created that combines the $CO_2$ flux numbers from Table 2 with selected regional estimates from Figure 3/Table S1 and is helpful in facilitating synthesis of regional estimates with those from the global model (see the edits and new paragraphs in section 3.5).

**3 Recommendation**

I recommend to accept this manuscript after moderate revision. The core of the manuscript is solid and well described, but there is substantial potential for enhancing the impact of this paper by strengthening the discussion and conclusion sections at the expense of the background section.

**Response:** Thank you.

**4 Specific comments**

Abstract line 10: "not well constrained": It is unclear whether this is a statement about the formal uncertainty estimate ($\pm 80$ Tg C $yr^{-1}$ or whether this is a statement of confidence, i.e., how uncertain are the authors about this uncertainty statement.

**Response:** Both are true but the intended meaning is that the 160 Tg/yr is not well constrained. Revised by replacing "number" by "flux."

Abstract, general: The abstract is written in a relatively general and rather descriptive manner. It would benefit greatly from being more quantitative, and from making the link to the underlying processes.

**Response:** The abstract was revised accordingly.

page 3, lines 10 to 20: This set of clear definition is very helpful. Yet later on, when the different studies are compared and assessed, it is not always clear how the originally reported numbers were then rescaled/transformed to refer just to the EEZ.

**Response:** This was clarified throughout taking care to state when the shelves, EEZ or MARCATS regions are considered.

page 4, Figure 2 and text: The perspective taken here is one that is almost purely one-dimensional in lateral direction, whereas in reality, alongshore transports and the vertical structure of the exchange with the adjacent open ocean matter a great deal (see e.g., Frischknecht et al. 2018). This is not that relevant for the air-sea $CO_2$ flux, but matters greatly for any discussion about the nature and magnitude of the lateral exchange of inorganic and organic carbon within the ocean.

**Response:** This figure was removed. No assumption about the number of dimensions in the horizontal direction was made in the text – everything there is completely consistent with a 3D view. The reference to Frischknecht et al. (2018) was added.

pages 6-17, entire section 3: As mentioned above, it would be helpful to know how the numbers from the original studies were re-scaled or transformed into the numbers reported and discussed in this section.

**Response:** This was clarified in section 3.5 and a new figure was added to make this more obvious. We only converted units to g C/m2/yr where original studies were reporting in moles (for consistency throughout the review) and converted between area-specific and area-integrated fluxes by dividing or multiplying by area where the original studies provided this information. No attempts have been made to extrapolate outside of the reported area.

pages 6-17, entire section 3: It would help the reader if the individual sections were structured in a similar manner. While the air-sea $CO_2$ flux is clearly always in the center, the different sections treat the other elements, such as the processes explaining the fluxes, and in particular, the lateral fluxes of organic and inorganic carbon very differently.

**Response:** Sections 3.1 to 3.4 were revisited to ensure they all follow the same general structure. Section 3.4 was edited significantly to make it more consistent with the previous three subsections. Some uneven level of detail with respect to lateral fluxes is inevitable because in some regions there is less information available from published studies.

page 6, line 10: "The anthropogenic component of a given carbon flux is defined as the difference between its preindustrial and present-day fluxes." This definition of the anthropogenic carbon component differs in an important manner from that adopted for global ocean carbon studies (see e. g., McNeil and Matear (2013).) With this very broad definition, the authors include also changes induced by natural processes as "anthropogenic". In contrast, the open ocean community tends to define the anthropogenic carbon component as just this part of the flux/change in storage that occurs as a result of the increase in atmospheric $CO_2$. I am painfully aware that there is no generally accepted definition, but I would argue that the chosen, very broad definition of "anthropogenic" is in the end more confusing than illuminating. This is particularly the case since the Bourgeois et al. paper used to constrain the anthropogenic $CO_2$ component follows the more classical open ocean definition (see there comment on page 4171).

**Response:** The definition was removed. Instead, "the carbon that was added to the atmosphere by anthropogenic activities" was added as a qualifier to describe anthropogenic carbon. And the reference to McNeil and Matear (2013) was added to refer the reader to more detail if desired.

page 7, Figure 3: I very much like this figure. I suggest to color code the different estimates to emphasize the very different types of estimates. For example, by differentiating between model- and observation-based estimates. And then one could further differentiate between those that really represent a long-term mean flux, versus those that pertain to a single year only, etc.

**Response:** The figure was modified so that model estimates are shown in a different shade.

page 12, line 9: You may want to consider adding the Frischknecht et al. (2018) study here. They show that about a third of the organic matter produced in the first 100 km gets transported offshore.

**Response:** Done.

page 13, lines 24-27: This paragraph does not fit here. I recommend to move it to the trends section.

**Response:** Done.

page 14, section 3.5: I urge the authors to reconsider their decision to downplay their regional estimates that much (see my second major comment above).

**Response:** Agree, please see responses above.

page 15, figure 5 and text: I very much like this synthesis of the fluxes. But I also think that the authors need to caveat this in an important manner. First, the uncertainties are probably underestimated given the unrepresented error contribution from temporal variability and other systematic error sources. Second, this implicitly assumes some form of steady-state. And third, there are probably very large regional differences hidden beneath this continental summary. It is probably worth to open the Pandora box a bit with regard to the regional aspects here, since I would expect that it is primarily the Arctic/high latitude regions that contribute to this net offshore transport.

**Response:** Below are individual responses to the four points raised in this comment:

1) "the uncertainties are probably underestimated given the unrepresented error contribution from temporal variability and other systematic error sources." – for the reasons stated in the text, an error estimate of 50% is deemed reasonable. Nevertheless, it is also stated repeatedly in the manuscript that there is not much quantitative information for accurate error estimates.

2) "this implicitly assumes some form of steady-state" – Steady-state was not assumed. An increase in the coastal carbon storage is explicitly considered (see last paragraph of Section 3.5) and represented in the budget (see Figure 5; DIC accumulation).

3) "there are probably very large regional differences hidden beneath this continental summary" – Yes, there are regional differences. This is now discussed (see 3rd and 4th paragraph of section 3.5) and visually represented in the new Figure 4.

4) "It is probably worth to open the Pandora box a bit with regard to the regional aspects here, since I would expect that it is primarily the Arctic/high latitude regions that contribute to this net offshore transport." – Yes, this is now discussed (see 3rd and 4th paragraph of section 3.5) and visually represented in the new Figure 4.

page 17-18: Section 4: Relative to the previous section 3, this section could benefit from some better structuring. It is also not so clear what message the authors want to convey except that "things are inconclusive". I think that more is possible here.

**Response:** Agree. This section has been significantly reworked.

page 17, lines 20 through 23: "by definition, changes in total carbon fluxes imply changes in anthropogenic fluxes as well." See my comment above about the definition of anthropogenic $CO_2$. I suggest to reconsider this very broad definition of "anthropogenic $CO_2$".

**Response:** The definition was removed (see response above). The sentence was reworded accordingly.

page 18, line 6: "pCO2 has increased faster than in the open ocean due to the combined effects of atmospheric uptake..." What is the evidence for this process being important? Such a behavior is not

easy to achieve in a system that is generally outgassing - here, only a reduction in the surface residence time could achieve this. Thus, the much more important process explaining the higher than expected trend is ocean circulation and biology.

**Response:** This sentence is one of several that were deleted in the revision.

page 18, lines 10ff: "Some studies suggest that trends in the air-sea $pCO_2$ gradient ($\Delta pCO_2$) are indicative of a strengthening or weakening of the net CO2 uptake by shelf systems...". This is not quite correctly formulated. A trend in $\Delta pCO_2$ does not "indicate" a trend in the air-sea $CO_2$ flux, but actually "implies" a trend.

**Response:** Yes, agree. Changed to "imply."

page 18, lines 20 through 26: I don't agree with the line of arguments here (or at least how I understood them). The authors seem to suggest that because the anthropogenic trend cannot be detected before 30 years have passed, the trends by Laruelle are likely not real. In my opinion, the discussion needs to be structured much more carefully. First, the trends detected by Laruelle et al. (2018) cover a rather short period, and thus are much more likely the result of internal variability than the result of an anthropogenic (forced) trend (in climate). Second, the time of emergence described by McKinley et al. (2016) checks for the emergence of a forced climate change signal versus that driven by internal variability. This does not imply that there is no trend to be detected, for example, in the Delta pCO2, if this trend was solely caused by the increase in atmospheric CO2, i.e., the uptake of anthropogenic $CO_2$ (sensu strictu). Please reconsider the writing of this paragraph.

**Response:** Agree, this paragraph conflated two different things. The text about time of emergence has been removed because it is not directly relevant for the questions discussed here.

page 19, lines 4 and 5: I don't know of any occasion where the impact of the changes in wind-stress on the gas transfer velocity had been more important than the impact on the ocean circulation. Thus, this sequence, i.e. gas exchange first, circulation second, does not reflect the common situation.

**Response:** The order here did not imply order of importance. The sentence has been reworded to make clear that the first effect is direct but the second, indirect effect is more important.

page 19ff, section 4.2: As section 4.1, this section could benefit from some better structuring. It largely consists of some general statements about ocean acidification, but does not really advance the knowledge much. It would greatly benefit from some more quantitative assessments of the distribution and the trends in OA.

**Response:** This section was edited significantly and published trends are synthesized.

page 20, line 4: "...biological production and respiration". Even more important is ocean transport and mixing. (see the next sentence)

**Response:** Added "ocean transport processes"

page 20, line 19: "events". Here is one example of why a better structuring of this section is needed. This sentence starts without context, and as a reader one is baffled about the expression "events", when the subject is long-term trend. etc.

**Response:** The previous sentence states that trends in coastal regions are more variable and was intended to set the stage. The phrase "and often event-driven" was added to make this more clear.

page 20, line 21, "Polar regions are naturally prone to acidification because of their low temperatures" This sentence is problematic for two reasons. First, the polar regions are not really prone to acidification per se (there sensitivity to change is actually lower, e.g., del pH/del pCO2), but what makes them prone is them being much closer to critical thresholds. Second, this sentence implies that their low pH or low Omega is a result of the waters being cold. This is not really correct either. It is because these waters have a high DIC/Alk ratio. And yes, the low temperatures are one of the reasons for the high DIC/Alk ratios, but also the freshwater inputs, the much more common mixing with deeper high DIC/Alk waters (owing to remineralization), and their generally low Alk are important factors. This is readily illustrated if one contrasts the pH and Omega values in the Atlantic to the Pacific at the same temperature.

**Response:** These sentences were corrected accordingly.

page 21, lines 34ff: "The main contributing factor to the relatively high rates of acidification in polar waters is retreating sea ice, which adds meltwater from multiyear ice and increases the surface area of open water, thereby enhancing the uptake of atmospheric CO2 (Cai et al., 2010a; Steiner et al., 2013)." This sentence is another good example for the need to better structure this section. It makes a statement about the cause of the faster trend for OA in the Arctic even before a statement has been made that the trends are actually larger. Thus, across the board in this section, it behooves the authors well to first establish a good base line of the expected trends, then report the observed trends and only then start to discuss the potential drivers for these trends.

**Response:** Yes, thank you. This paragraph was edited significantly.

page 21, lines 21ff: "more prone". See my comment above. It would be good to be more specific what is meant here.

**Response:** Agree, this was edited accordingly.

page 21, Conclusions. Before advancing to the conclusion section, it would strengthen the manuscript substantially if the authors expanded the discussion section by considering additional material (see my first major comment above). Examples, are coastal versus open ocean, comparison of fluxes across compartments, and recommendation for future studies, etc.

**Response:** The conclusions were strengthened by adding more quantitative information.

page 22: line 9-10: "Ultimately, the removal of anthropogenic carbon is the relevant quantity for assessing the contribution of ocean margins to the uptake of anthropogenic carbon". This is confusing - what is the difference between "removal" and "uptake"? Please explain better what is meant here. Is it global versus coastal?

**Response:** Changed to "the uptake of anthropogenic carbon" – no difference was implied by using "removal" but "uptake" seems to be the better word choice.

**5 Cited References**

McNeil, B. I., and R. J. Matear (2013), The non-steady state oceanic $CO_2$ signal: its importance, magnitude and a novel way to detect it, Biogeosciences, 10(4), 2219– 2228, doi:10.5194/bg-10-2219-2013.

Frischknecht, M., M. Münnich, and N. Gruber (2018), Origin, Transformation, and Fate: The Three-Dimensional Biological Pump in the California Current System, J. Geophys. Res. Ocean., 123(11), 7939–7962, doi:10.1029/2018JC013934.

**Response to the interactive comment by Reviewer 2 (Christophe Rabouille)**

Below the entire review is pasted in black and responses are in blue font.

The paper presents a synthesis of carbon cycling on the continental margins of the United States , including the Atlantic, the Pacific and the Arctic coasts. The authors concentrate on the CO2 exchange between air and seawater as it is the most documented and tabulated parameter for carbon exchange in the ocean, and specifically in coastal zones. They propose an overall map of air-sea CO2 exchange for the East, West and Arctic coasts which shows a large tendency of these shelves to be a sink for CO2 (Fig. 3) and then discuss the results by geographic zone which makes a long and rather tedious paper to read. At the end, the acidification question is raised and shortly discussed. Overall, the paper is long but well written and proposes a good synthesis for all researcher interested by carbon cycling in the coastal zone. By discussing the different regions separately and addressing different "cases" (large shelves, upwelling, arctic, enclosed sea-GoM), they provide insightful explanation for the observed CO2 fluxes. I think that the paper is publishable after minor revisions.

**Response:** We would like to thank Christophe Rabouille for reviewing our manuscript and for providing his constructive and positive comments. Addressing these and Reviewer 1's comments resulted in a much-improved presentation. Point-by-point responses and resulting changes are given below.

Detailed comments:

Page 3, line 10: The coastal ocean is defined as "non-estuarine waters", which is questionable regarding CO2 budgets for the coastal zone since estuaries and deltas are sources of similar magnitude compared to coastal ocean sink. . . It should be emphasized at this point in text that this cutting off estuaries and deltas is because of the paper by Najjar et al. (2018) which has already addressed the estuarine and tidal wetland's part of the C Cycle.

**Response:** The following sentence was added in section 1 to explain why estuarine waters are not included. "*The review does not consider estuarine waters and tidal wetlands as these are the subject of a separate review (Windham-Myers et al. 2018).*"

Page 5, line 15: "colder shelf water is denser". The authors should cite reference work by Canals et al. (2006, Nature Vol 444, doi:10.1038/nature05271) on cascading in canyons which is the ultimate case of cold and dense water diving in the lighter open sea.

**Response:** This reference was added and the Ivanov et al. (2004).

Ivanov, V. V., Shapiro, G. I., Huthnance, J. M., Aleynik, D. L. & Golovin, P. N. Cascades of dense water around the world ocean. Progr. Oceanogr. 60, 47–98 (2004).

Page 6, line 8: "burial and export of carbon. . . remove atmospheric CO2". The authors should specify that this is true only for ORGANIC carbon, as inorganic carbon burial will result in net CO2 increase in the zone of CaCO3 formation (2 HCO3- + Ca2+ → CaCO3 + CO2 + H2O).

**Response:** Changed to "*burial of organic carbon.*"

Page 14, line17-18: "DIC is a component of the brine whereas total alkalinity precipitates. . .as ikaite". I do not understand why alkalinity would precipitate without DIC. He sentence should be modified to

"precipitation of CaCO3 as ikaite changes the DIC to Alkalinity ratio by consuming two times more Alk than DIC during the process. . ."

**Response:** This paragraph has been rewritten.

Page 18, line 10: ". . . pCO2 gradient are indicative" I would rather say "is the appropriate indicator for"

**Response:** Changed simply to "indicate."

Page 18, line 12: "and cross shelf exchange", please add "potential" cross shelf exchange

**Response:** Adding "potential" would change the meaning in a way that is not intended.

Page 19, Fig. 6 legend: please add uncertainty to the slopes "1.86 ppm/y and 1.95 ppm/y " as reported in text for the shelf seas (pge 18, line 14-17).

**Response:** Done.

Page 20, line 10: cut sentence after "oyster larvaes in the coastal Pacific Ocean".

**Response:** Done.

Page 22, line 13-16: These sentences which describe the overall CO2 flux in the coastal ocean make little sense. The main conclusion of the study is that these margins are sinks for CO2 (either natural or anthropogenic) and this should be the first and main sentence in this paragraph, before summarizing the details.

**Response:** These sentences were removed and the paragraph revised by adding more quantitative budget information.

---

## Author Response (AR2)

Dear Jack,

Thank you very much for handling this manuscript and thanks you for pointing out the errors below.

-p.4, l.21 is unclear as written. ... in the absence of electron acceptors DIC and $CH_4$ are produced....

Response: Yes, this became garbled during editing. Changed to: "and, in the absence of electron acceptors other than $CO_2$, $CH_4$"

- p.12, l. 25.. Gulf differ markedly.....

Response: Corrected.

Kind regards,
Katja